# Transcriptomes and neurotransmitter profiles of classes of gustatory and somatosensory neurons in the geniculate ganglion

Gennady Dvoryanchikov [1], Damian Hernandez[1], Jennifer K. Roebber [2], David L. Hill[3], Stephen D. Roper [1,2,4] & Nirupa Chaudhari [1,2,4]

Taste buds are innervated by neurons whose cell bodies reside in cranial sensory ganglia. Studies on the functional properties and connectivity of these neurons are hindered by the lack of markers to define their molecular identities and classes. The mouse geniculate ganglion contains chemosensory neurons innervating lingual and palatal taste buds and somatosensory neurons innervating the pinna. Here, we report single cell RNA sequencing of geniculate ganglion neurons. Using unbiased transcriptome analyses, we show a pronounced separation between two major clusters which, by anterograde labeling, correspond to gustatory and somatosensory neurons. Among the gustatory neurons, three subclusters are present, each with its own complement of transcription factors and neurotransmitter response profiles. The smallest subcluster expresses both gustatory- and mechanosensory-related genes, suggesting a novel type of sensory neuron. We identify several markers to help dissect the functional distinctions among gustatory neurons and address questions regarding target interactions and taste coding.

[1] Department of Physiology & Biophysics, University of Miami Miller School of Medicine, Miami, FL 33136, USA. [2] Graduate Program in Neurosciences, University of Miami Miller School of Medicine, Miami, FL 33136, USA. [3] Department of Psychology, University of Virginia, Charlottesville, VA 22904, USA. [4] Department of Otolaryngology, University of Miami Miller School of Medicine, Miami, FL 33136, USA. Gennady Dvoryanchikov and Damian Hernandez contributed equally to the work. Correspondence and requests for materials should be addressed to N.C. (email: nchaudhari@med.miami.edu)

The cells of mammalian taste buds have been extensively studied. We know many of the membrane receptors and transduction and processing pathways that mediate detection of taste stimuli[1, 2]. Yet, we have limited understanding of the cranial sensory ganglion neurons that innervate taste buds. Evidence of the diversity of these neurons comes from electrophysiology[3, 4], Ca2+ imaging[5, 6], and analyses of neurotrophin dependence[7, 8]. However, the lack of fundamental molecular information impedes progress towards an integrated framework including the specificity of synaptic contacts with taste bud cells, the identities of the circuits that process taste, and the anatomical underpinnings for taste coding.

Molecular characterization of afferent neurons has yielded key information in other sensory systems. For example, single-cell RNA sequencing has greatly refined our understanding of how neurons in dorsal root ganglia parse and transmit distinct modalities of somatosensory stimuli[9, 10]. The benefits that accrue from this approach for all areas of the central nervous system (CNS) and peripheral nervous system (PNS) are many, and include the deployment of fluorescent reporters for functional studies and markers to facilitate neuroanatomical and connectome analyses[9, 11, 12]. Detailed molecular characterizations may also identify novel cell types and circuits that encode sensory submodalities[11] and discover sites of developmental or physiological regulation[12, 13].

To begin remedying this major gap in our understanding of taste, we conducted single-cell RNA sequencing of neurons from the mouse geniculate ganglion, the cranial ganglion that innervates taste buds on the anterior tongue and palate. Our unbiased transcriptome analyses of single geniculate ganglion neurons reveal distinctive clusters of cells that are characterized by specific markers, paving a way to discover the specificity of the synaptic contacts with peripheral receptors and central relays and the significance of these contacts for sensory coding.

## Results

**Neuron capture and sequencing.** To understand the cellular diversity of neurons that innervate taste buds, we acquired single-cell transcriptome data on gustatory afferent neurons. In mammals, geniculate and petrosal ganglia innervate palatal and lingual taste buds. We focused on the geniculate ganglion because a large fraction of its neurons are gustatory (Fig. 1a, b) and because the majority of neurophysiological recordings and behavioral analyses are based on activity in this ganglion. The small size of the mouse geniculate ganglion ($\approx$ 800–900 neurons[14]) and its location, surrounded by bone at the base of the cranium, preclude methods such as drop-seq which sample tens

of thousands of cells. Instead, we selected the Fluidigm C1 system which permits capturing individual neurons starting with only a few thousand cells, while affording high sensitivity and sequence coverage[15]. Geniculate ganglia from C57BL6/J mice were dissociated, the suspension was depleted of glia, and neurons were captured in C1 microfluidic chips (Fig. 1c, Methods and Supplementary Fig. 1, 2a–e). Single neurons that met stringent criteria (Supplementary Fig. 2f) were used to produce individually barcoded libraries and subjected to RNAseq in a single lane. The 96 neurons sequenced in the present study represent >10% of the neurons in a geniculate ganglion, and thus are likely to include all but the most infrequent neuronal subtypes.

All cells were sequenced to a depth of at least $10^6$ mapped reads per cell. We detected expression of ≥13,000 genes from every cell (Supplementary Fig. 3), which compares favorably with most published reports on neurons[9, 12, 15, 16].

We began our analysis by identifying the most highly expressed genes (Fragments Per Kilobase per Million mapped reads (FPKM) ≥200 across the entire set of 96 cells). Expression levels for these 333 genes across all 96 cells were subjected to unsupervised hierarchical clustering analysis (HCA, Fig. 2a). The resulting dendrogram and correlogram of Pearson Correlation Coefficients clearly displayed two major groupings, with 59 (yellow) and 37 (black) cells, respectively. To get a broader and independent view of gene expression while reducing the dimensionality of the data set, we conducted Principal Component Analysis (PCA) of all (>17,000) expressed genes across all 96 cells. PCA also yielded primarily two groupings (Fig. 2b). Importantly, the 96 cells assorted into the same groupings (yellow and black) from these two independent analyses, which respectively were based on 333 (HCA) or ≥17,000 (PCA) genes (Fig. 2a, b).

**Taste versus somatosensory.** To assess the significance of these two clusters of neurons, we examined the genes that contributed substantial variance and were expressed differentially between the two clusters. *P2rx2* and *P2rx3*, encoding ionotropic purinoceptors that function in afferent transmission from taste buds to nerves, were limited to neurons in the yellow cluster. Further, the transcription factor, *Phox2b*, which has been associated with specification of gustatory fate during development[17] was found only in the yellow cluster. In contrast, we noted that neurons in the black cluster selectively expressed several genes that have previously been associated with somatosensory neurons. Three examples are shown (Fig. 2c). They include *Fxyd2*, a subunit of Na-K-ATPase that is associated with the maturation of peptidergic nociceptors[18] and *Prrxl1* (also called *Drg11*), a transcription factor essential for

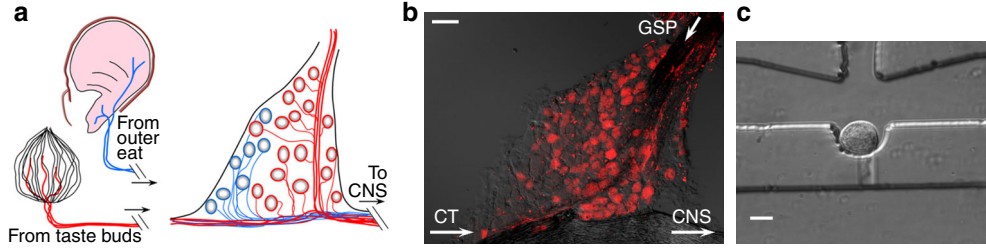

**Fig. 1** Peripheral targets innervated by geniculate ganglion sensory neurons. **a**. Gustatory afferent fibers from taste buds in anterior tongue and palate (red) and somatosensory fibers innervating the pinna (blue) converge in the geniculate ganglion where their somata reside. **b**. Cryosection of a geniculate ganglion following anterograde labeling of gustatory nerves (GSP, CT) with tetramethylrhodamine dextran. Taste neurons are red while somatosensory neurons in the ganglion remain unstained. *Arrows* indicate direction of propagation of sensory signals from peripheral targets. **c**. A geniculate ganglion neuron in a Fluidigm chip capture site. Several ganglia were dissociated and processed for cell capture (see Methods). 96 such singly captured neurons were sequenced to yield data on neuronal cell types. *Scale bars*, **b**, 50 μm **c**, 20 μm

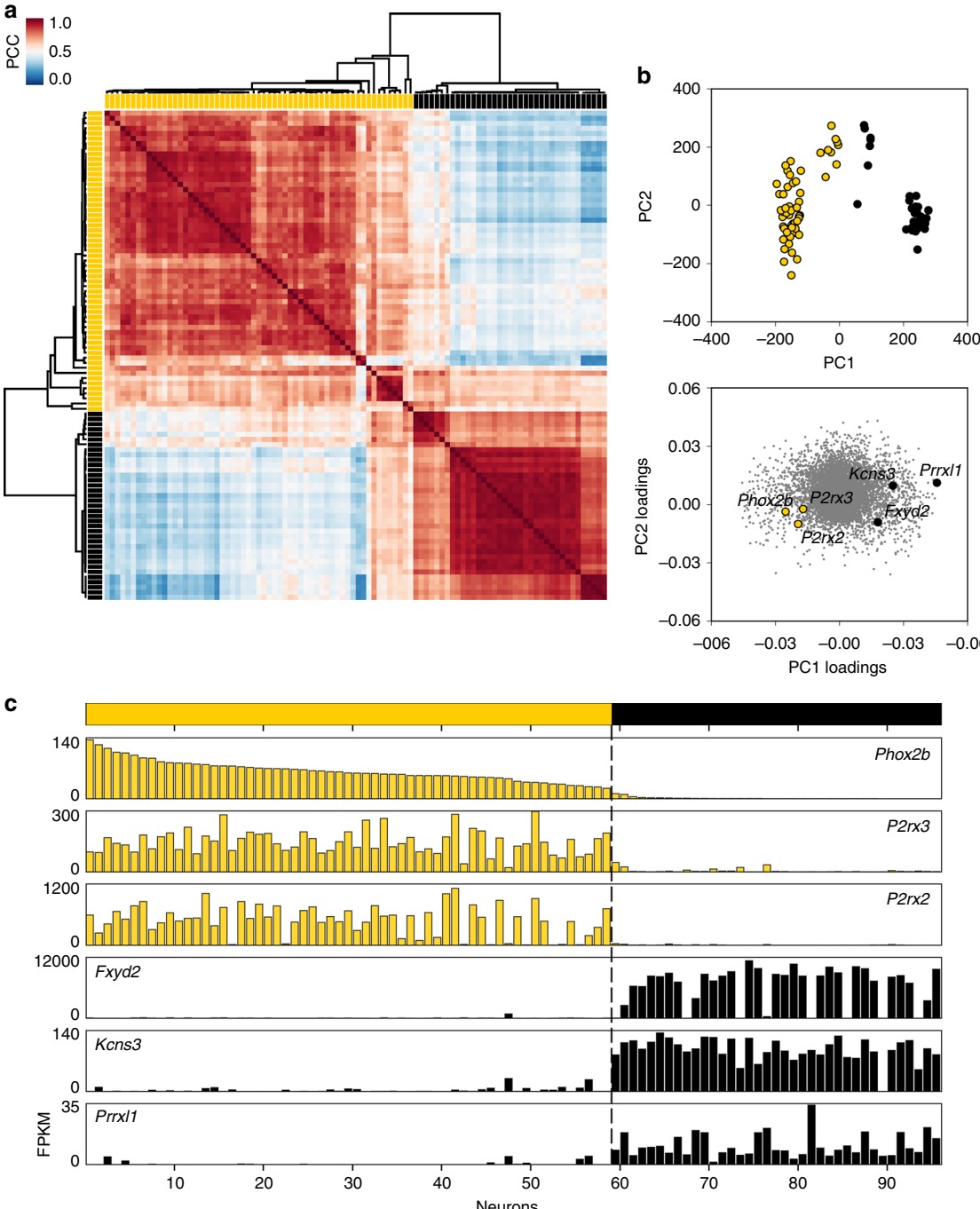

**Fig. 2** Geniculate ganglion neurons belong to two main groups, gustatory and somatosensory. **a**. Correlogram and dendrogram resulting from unsupervised hierarchical clustering analysis of 96 geniculate ganglion neurons. Here, genes were selected that displayed average FPKM values ≥ 200 across all 96 cells (i.e. the 333 most highly-expressed genes). The correlogram displays Pearson Correlation Coefficient (PCC) values for each cell against all others on a heat map scale. The dendrogram shown at *top* and *left* reveals two major groupings (59 and 37 neurons) which are assigned yellow and black labels, respectively. **b**. Principal component analysis (PCA) of all 96 geniculate ganglion neurons, based on all 17,225 genes expressed. The *upper* plot displays a symbol for each of the 96 cells according to their scores on the first two eigenvectors (PC1, PC2) which account for most variance across the set. Individual cells are color-coded yellow and black as in **a**. It is apparent that cells separate into two principal groupings and that these correspond to the groupings produced by the hierarchical clustering of a much smaller number of genes shown in **a**. The lower plot shows the PC1 and PC2 loadings, with each detected gene represented by a *grey* symbol. Genes previously associated with gustatory neurons, *Phox2b*, *P2rx2*, and *P2rx3* are annotated (yellow) and display similar loadings. Examples of genes which were highly-expressed and are selective for the black group (*Fxyd2*, *Kcns3*, and *Prrxl1*) are also indicated. **c**. Bar graphs of FPKM values for six group-selective genes identified in **b**. The neurons are sorted by FPKM values for *Phox2b*

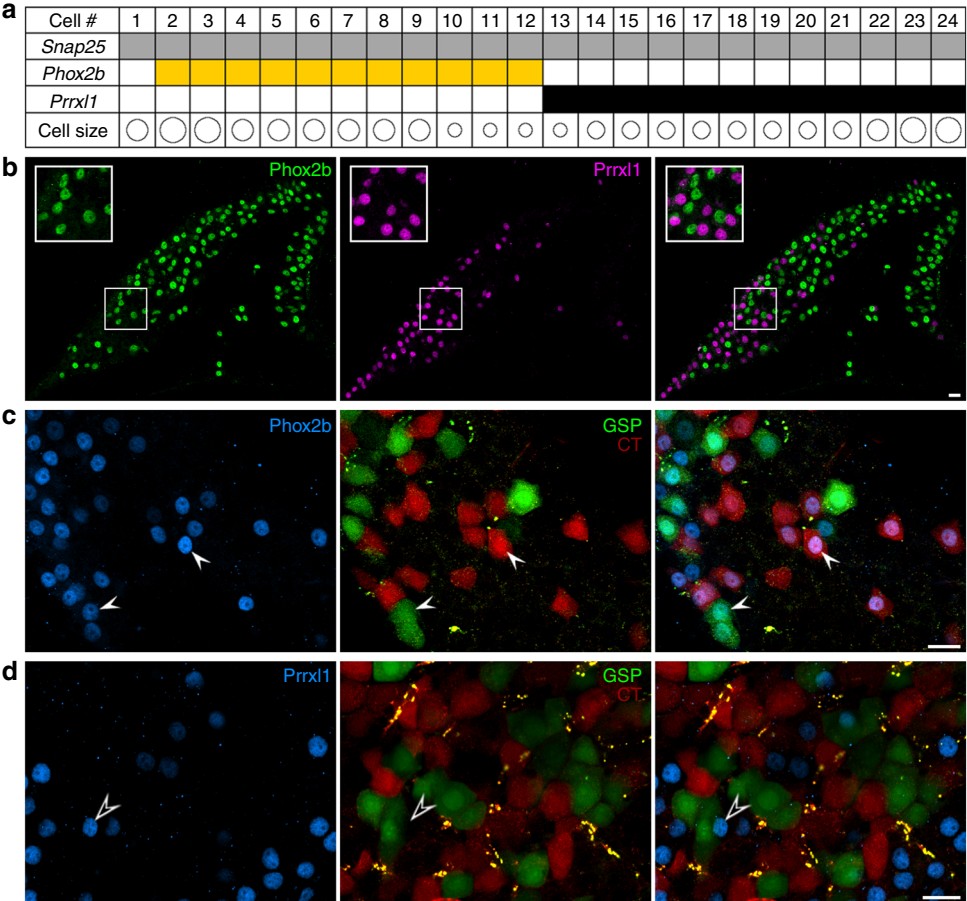

**Fig. 3** Phox2b and Prrxl1 are markers of gustatory and somatosensory neurons, respectively. **a**. RT-PCR of single geniculate ganglion neurons confirms that Phox2b and Prrxl1 are expressed in non-overlapping sets of neurons. 24 freshly isolated geniculate ganglion neurons were analyzed by single-cell RT-PCR for *Snap25* (positive control), and two test genes. Product from the end-point PCRs is indicated by color in the grid corresponding to each cell. The transcription factor, *Phox2b* (yellow), previously associated with developing taste neurons, was detected in about half the neurons (11 of 24), but never with *Prrxl1* (black). The size of each neuronal soma was observed before harvesting and is depicted schematically in the bottom row. Both groups included neurons of various sizes (18–35 μm diameter). **b**. Cryosection of geniculate ganglion, immunostained for Phox2b (green) and Prrxl1 (magenta) shows that the two transcription factors define non-overlapping neuronal populations. Inset in each panel is an enlarged view of the boxed area of the ganglion. Only 4 of 1453 nuclei stained for both Phox2b and Prrxl1; 1 nucleus stained for neither (3 mice). **c**. Gustatory neurons, visualized by anterograde labeling of chorda tympani with TRITC-dextran and greater superficial petrosal with FITC-dextran. Of 537 anterograde labeled neurons, 501 (93%) contained a Phox2b+ nucleus (3 mice). The remaining 7% of neurons can be accounted for as partial profiles that included labeled cytoplasm but lacked the nucleus (because the nucleus occupies only ≈ 80% of the diameter of each cell). **d**. Gustatory-labeled neurons very seldom included nuclei that immunostained for Prrxl1 (4 of 321 neurons counted in 9 non-adjacent sections from 3 mice). *Scale bars*, 20μm

the formation of certain nociceptors[19, 20] and expressed in dorsal root and trigeminal ganglia. These initial observations led us to hypothesize that the yellow and black clusters represent gustatory and somatosensory neurons respectively.

We used three approaches to test this hypothesis. First, we validated the RNAseq observation by conducting single-cell RT-PCR on 24 freshly isolated geniculate ganglion neurons. Aliquots of each single cell cDNA were tested for *Snap25* as a positive control and the two transcription factors, *Phox2b* and *Prrxl1*. Second, we used double immunofluorescence microscopy to visualize the corresponding proteins. Both methods demonstrated that *Phox2b* and *Prrxl1* display a mutually exclusive pattern of expression in geniculate ganglion neurons (Fig. 3a, b). Further, Phox2b and Prrxl1 immunoreactive nuclei accounted for virtually all neurons in the ganglion. The Prrxl1-stained nuclei (somatosensory neurons) were found predominantly at one pole (basal-peripheral) of the triangular ganglion (Fig. 3b). Third and most importantly, we tested the hypothesis by anterograde labeling of the two taste nerves that enter this ganglion, using different fluorescent dextrans applied to the chorda tympani (CT) and the greater superficial petrosal (GSP)[21]. Nearly every neuron that was anterograde-labeled on taste afferents possessed a nucleus immunopositive for Phox2b (Fig. 3c). Conversely, anterograde-labeled taste neurons seldom included a Prrxl1-immunoreactive nucleus (Fig. 3d).

We also observed that *Pou4f1* and *Pou4f2* (also called *Brn3a*, *3b*), previously associated with the developmental specification of a somatosensory fate[17], are selectively expressed in the black cluster neurons (Supplementary Fig. 4a, c, d) and can be detected by immunostaining (Supplementary Fig. 4e) in a somatosensory pattern along the long edge of the ganglion, similar to Prrxl1 (as in Fig. 3b).

In combination, these findings identify the presence of the two main classes of neurons in the geniculate ganglion, validate them as gustatory and somatosensory, and identify Phox2b and Prrxl1 respectively as robust markers for these major clusters.

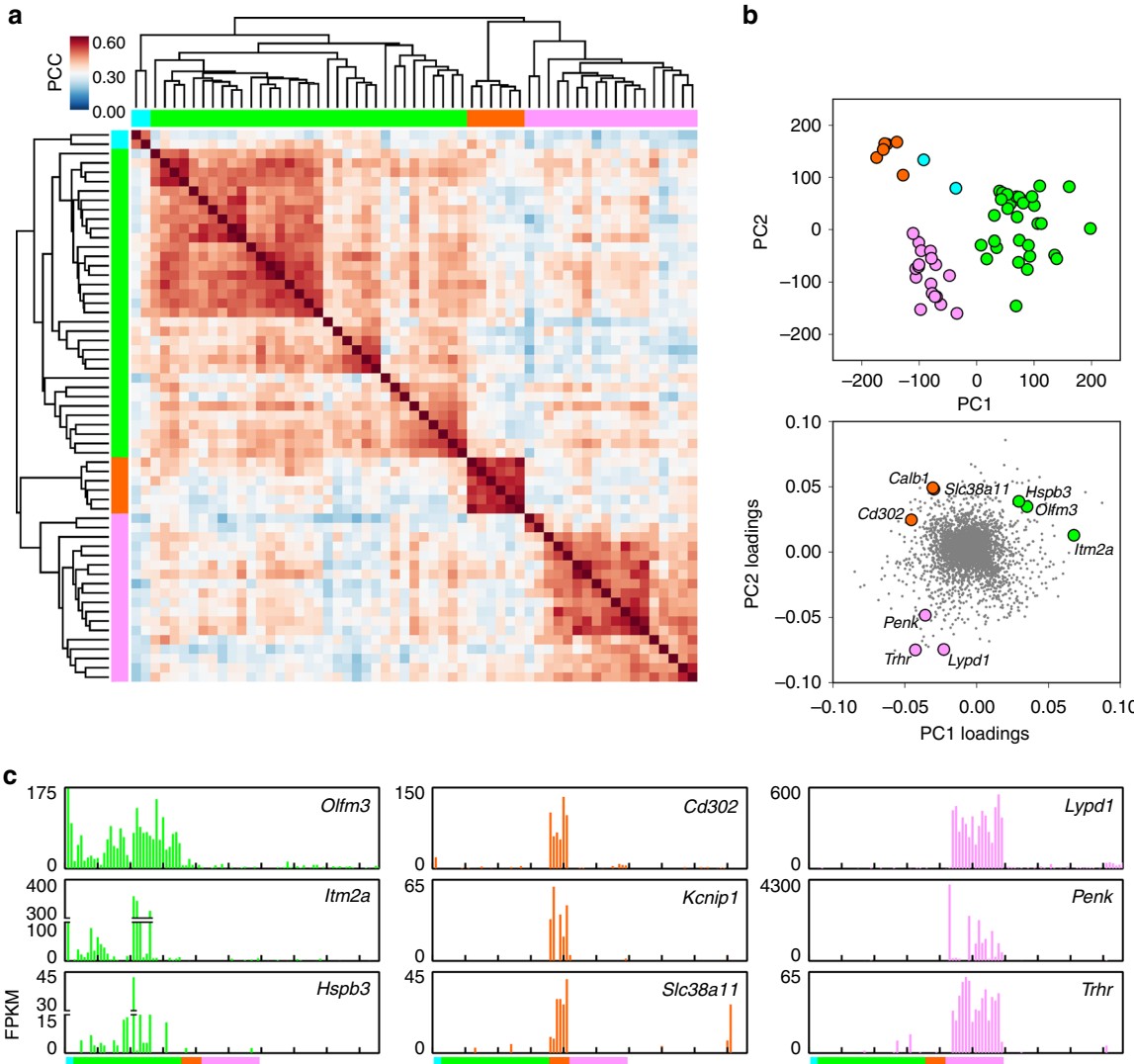

**Fig. 4** Gustatory neurons in the geniculate ganglion fall into three sub-clusters. **a**. The 59 sequenced neurons identified as gustatory (Fig. 2a) were subjected to hierarchical clustering of Pearson Correlation Coefficients based on 580 taste neuron-selective genes (FPKM values ≥ 5-fold higher in gustatory neurons and FPKM ≥ 1, averaged across all 59 gustatory neurons). The dendrogram across the *top* and *left* indicate three main sub-clusters, which are assigned colors, T1 green (33 neurons), T2 orange (6 neurons), T3 pink (18 neurons). Two neurons (cyan) at far left do not cluster well with any others. The correlogram displays PCC values for each cell against all others on a heat scale. **b**. Principal component analysis of the same 59 gustatory neurons, assessed for 8150 genes expressed at significant levels in taste neurons, plotted according to PC1 and PC2 scores. Individual cells are color-coded as indicated by hierarchical clustering in **a**. Loadings of the 8150 genes from PC1 and PC2 are plotted in the *lower* scatter plot. Three representative genes selective for each gustatory subcluster are indicated by symbols colored as in **a**. **c**. *Bar graph* of FPKM values for 3 genes that are selectively expressed in each of the 3 gustatory sub-clusters. 59 gustatory neurons are arrayed according to the subgroups assigned by the dendrogram (colors as in **a**), followed by 37 somatosensory neurons (black, as in Fig. 2a)

**Taste neuron types**. Next, we asked if the 59 neurons in the taste (yellow) cluster could be further classified into sub-clusters. For this, we filtered the data to identify 580 most taste-selective genes (FPKM ≥ 5-fold higher in taste relative to somatosensory neurons, see Methods). These 580 genes were subjected to a second iterative hierarchical clustering analysis (Fig. 4a). The dendrogram revealed three major sub-clusters, with 33 (green, T1), 6 (orange, T2) and 20 (pink, T3) neurons, respectively. Two additional neurons (cyan) at the extreme left may represent additional, infrequent cell type(s) (Fig. 4a). We also conducted PCA on the 59 taste neurons for all 8150 genes that are significantly expressed (FPKM ≥ 10) in at least 3 taste neurons (see Methods). Importantly, the groupings revealed by HCA of differentially expressed genes were maintained in the independent PCA of a much larger set of genes (Fig. 4b). That is, two independent

analyses sorted gustatory neurons into similar groupings (T1, T2, T3). Gene loadings for this PCA (Fig. 4b, lower) suggested that numerous genes are preferentially expressed in each of the taste sub-clusters. Expression levels for three exemplar genes for each sub-cluster are presented for each of the 96 cells (Fig. 4c). These exemplar genes encode proteins spanning a diversity of neuronal functions, including neuropeptides (*Penk*), receptors and transporters (*Slc38a11*, *Trhr*), cell surface adhesion and recognition molecules (*Olfm3*, *Cd302*), and proteins related to cellular metabolism and trafficking (*Itm2a*, *Hspb3*, *Kcnip1*, *Lypd1*).

**Transcription factors**. If the gustatory sub-clusters represent distinct neuronal cell types, they should express unique profiles of transcription factors that determine their identities. Thus, we conducted PCA on all 59 gustatory neurons based on their

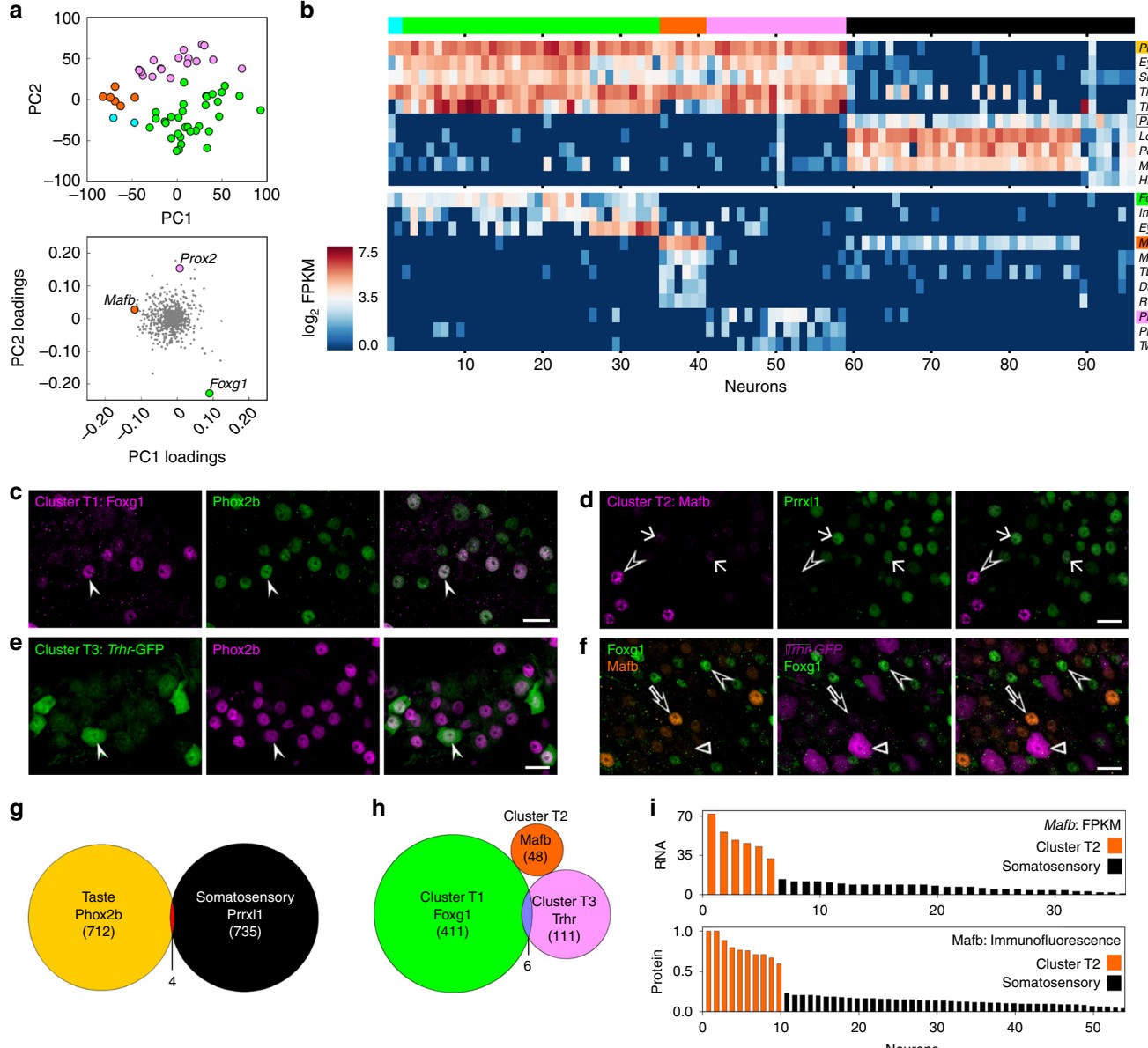

**Fig. 5** Transcription factors (TFs) define non-overlapping subclusters of gustatory neurons. **a**. PCA of 59 gustatory geniculate ganglion neurons, colored as in Fig. 4a, based on expression of 976 TFs (*upper* plot). Principal component loadings (*bottom* plot) reveal divergent positions occupied by sub-cluster-selective TFs. **b**. Expression levels (log$_2$ FPKM) of TFs in all 96 neurons. Gustatory neuron sub-clusters, T1, T2, T3, are green, orange and pink respectively; somatosensory neurons are black. The *upper* heat map shows expression of 5 gustatory- and 5 somatosensory-selective TFs. The *lower* heat map shows expression levels for 11 gustatory sub-cluster-selective TFs. **c**. Immunostaining of a geniculate ganglion for Foxg1 (proposed T1 marker) and Phox2b (gustatory marker). All Foxg1+ neurons counted (243 neurons; 3 mice) were also Phox2b-immunoreactive filled arrowhead; Foxg1+ neurons accounted for 59% of Phox2b+ neurons. **d**. Immunostaining of a geniculate ganglion with anti-Mafb (proposed T2 marker) and anti-Prrxl1 (somatosensory marker). Mafb-bright nuclei (unfilled triangle) do not overlap with Prrxl1-immunoreactivity (0 of 29 nuclei; 3 mice). However, faint staining for Mafb was also detected in many Prrxl1-positive (i.e. somatosensory) nuclei. **e**. Immunostaining of a ganglion from a Trhr-GFP mouse (proposed T3 marker) with anti-GFP and anti-Phox2b. 80 of 81 GFP+ neurons (3 mice) contained a Phox2b-immunoreactive nucleus. **f**. Triple immunostaining for Foxg1, Mafb, and GFP on geniculate ganglia from a *Trhr*-GFP mouse reveals a mutually exclusive pattern of expression of the three markers. **g**. Venn diagram of Phox2b, Prrxl1, and NeuN expression based on triple immunofluorescence (expanding on Fig. 3b). Of 1452 NeuN+ nuclei examined (from 3 mice), only 4 were immunoreactive for both Phox2b and Prrxl1 and 1 stained for neither. **h**. Venn diagram for taste sub-clusters, based on triple immunofluorescence for Mafb, Foxg1 and GFP on geniculate ganglia from Trhr-GFP mice. 570 of 576 cells (3 mice) displayed a non-overlapping pattern of the three markers. **i**. *Bar graphs* compare FPKM values for *Mafb* (T2 neurons) across all sequenced neurons and quantified fluorescence after immunostaining for Mafb. Co-staining with Prrxl1 (as in **d**) was used to assign neuronal identity to either taste cluster T2 or the somatosensory cluster. All *scale bars*, 20μm

expression levels across a list of 976 transcription factors, co-factors and chromatin-remodeling proteins (collectively termed TFs). When color codes designating the three taste neuron sub-clusters (Fig. 4a) are displayed for all 59 neurons in the

TF-based PCA, it is apparent that neurons within a taste sub-cluster have a similar pattern of TF expression (Fig. 5a). It should be noted that fewer than 8% of the 580 genes used in the initial HCA (Fig. 4a) were TFs. That is, the two independent analyses,

HCA and PCA, were conducted on substantially different sets of genes, and produce identical outcomes of cellular groupings. The gene loadings in the PCA (Fig. 5a) suggest that many candidate TFs are expressed selectively in each taste sub-cluster. To visualize these, we generated a heat map display (Fig. 5b) of FPKM values across all 96 cells for gustatory, somatosensory and sub-cluster selective TFs.

We then validated the patterns of TF expression using immunostaining. We selected candidates based on disparate positions on the gene loadings plot (Fig. 5a) and by their computed differential expression across taste sub-clusters (Fig. 5b). These analyses suggested *Foxg1*, a forkhead box family TF, as a marker for sub-cluster T1 (green). Antibody against Foxg1 stained a large subset of neuronal nuclei in geniculate ganglion (Fig. 5c) and all were also immunoreactive for *Phox2b*. This confirmed that *Foxg1* is limited to a subset of taste neurons. For sub-cluster T2 (orange), we selected *Mafb*, a bZIP transcription factor. We stained sections of geniculate ganglia with antibodies against Mafb and Prrxl1, the somatosensory marker. Only a few nuclei stained brightly for Mafb (Fig. 5d, unfilled triangle), consistent with the small population of neurons in the T2 sub-cluster. All such Mafb-bright nuclei were negative for Prrxl1 (Fig. 5d). In addition, we observed many neuronal nuclei that stained only faintly for Mafb, and were immunoreactive for Prrxl1 (arrow, Fig. 5d). In RNAseq data, we noted that most somatosensory neurons also expressed *Mafb*, but at FPKM values 5–10 fold lower than in taste sub-cluster T2 (Fig. 5i). To compare RNA and protein distributions, we quantified Mafb immunofluorescence. At both RNA and protein levels (Fig. 5i), Mafb displays strong expression in sub-cluster T2, weak expression in somatosensory, and no expression in other neurons. This observation was also confirmed by double staining for Mafb and Phox2b (Supplementary Fig. 6).

For taste sub-cluster T3 (pink), we considered several candidate TFs suggested in the heat map. However, low expression levels and lack of highly specific antibodies prevented completion. Instead, we used a surrogate marker, *Trhr*, that is selectively expressed in all sub-cluster T3 neurons (Fig. 4c). We visualized *Trhr* expression via a transgenic strain, *Trhr*-GFP (Supplementary Fig. 5c, d). Geniculate ganglia from *Trhr*-GFP mice contained a few distinctly labeled neurons that were always Phox2b-immunoreactive (i.e. gustatory neurons; Fig. 5e, filled arrowhead). We also stained *Trhr*-GFP ganglia for both Foxg1 and Mafb (Fig. 5f) and found that neurons only express one of the three markers, confirming that these three markers, Foxg1, Mafb and Trhr, define three non-overlapping sets of gustatory neurons in the ganglion (Fig. 5h). It was not possible, given the constraints of combining primary antibodies, to test whether the T1, T2 and T3 sub-clusters account for all taste neurons.

The dendrogram that identifies taste sub-clusters (Fig. 4a) suggests that T1 and T3 may include further sub-classes. However, the 59 gustatory neurons in this analysis do not offer sufficient power to resolve these groupings at present.

**Neurotransmitter phenotype**. To assess if geniculate ganglion neurons differ in the neurotransmitters to which they respond, and if this sensitivity varies according to the groupings we observed above, we examined receptors for three neurotransmitters previously associated with taste bud cells. ATP is the documented transmitter and P2rx2/3 are the cognate receptors at the peripheral afferent synapse[22]. 5HT and GABA are synthesized in and secreted by one or more cell types in mammalian taste buds[23–25]. Thus, we screened our sequence data to identify the most prominently expressed receptors for each of these three transmitters (Fig. 6a). Among the P2 receptors, *P2rx2/3* displayed

the highest FPKM values and expression was limited to taste neurons. Surprisingly, the 6 neurons in taste sub-cluster T2 (orange) expressed much lower levels of *P2rx2* than other taste neurons. Three metabotropic purinoceptors, *P2yr1*, *P2yr12* and *P2y14*, were also expressed in many of the 96 sequenced neurons, but at much lower levels per cell. Among receptors for 5HT, the ionotropic receptor gene, *Htr3a* was most prominently expressed as previously reported[26]. *Htr3a* was seen consistently in sub-cluster T3 (FPKM 88-1074), but also in many additional neurons (FPKM up to 485). *Htr3b* was expressed primarily in somatosensory neurons (black cluster), along with the metabotropic receptor, *Htr1d*. Among receptors for GABA, the ionotropic receptor essential subunits, *Gabra1* and/or *Gabra2*, were detected in all sequenced neurons with higher levels in taste neurons. The β, γ and δ subunits required for building functional GABA receptors and conferring diverse pharmacological properties also were expressed across the 96 neurons in the RNAseq data set and confirmed by single-cell RT-PCR (Supplementary Fig. 7a, b).

We validated RNAseq data on the most prominently expressed subunits using single-cell RT-qPCR on 38 freshly isolated geniculate ganglion neurons. Each cell's cDNA was tested for *Snap25* (positive control), *P2rx2*, *P2rx3*, *Htr3a* and *Gabra1* (Fig. 6b). The data permitted us to classify individual neurons, *post-hoc*, into the sub-clusters established above. For example, neurons lacking both *P2rx2* and *P2rx3* are similar to the somatosensory (black) cluster, cells with strong expression of *P2rx2/3* but without *Htr3a* are similar to sub-cluster T1 (green) cells, while cells expressing *P2rx3* without *P2rx2* are similar to sub-cluster T2 (orange) cells.

For an independent validation of sequencing data, we employed immunofluorescence for P2rx2 and Gabra1 on ganglia from *Htr3a*-GFP mice[27]. First, we visualized native fluorescence of GFP without immunostaining to obtain a more quantitative estimate of *Htr3a* gene expression. We observed graded intensities of fluorescence, consistent with sequence and RT-PCR data. Many GFP-expressing neurons were immunoreactive for P2rx2 (Fig. 6c, ▲). Across the sections, neurons expressed GFP and/or P2rx2, i.e. in a pattern similar to sequence and RT-PCR data (Fig. 6a, b). Antibody against Gabra1 displayed staining in the majority of neurons, including some with and others without GFP (Fig. 6d).

Third, we tested for neurotransmitter receptor function using GCaMP3-expressing geniculate ganglion neurons[6]. Dissociated ganglion neurons were stimulated with KCl to confirm viability (see Methods and Supplementary Fig. 8b, c). Based on the expression of several purinoceptors in geniculate ganglion neurons, we tested for responses to ATP. To discriminate between P2rx- and P2ry-mediated signals, we tested whether responses were dependent on extracellular $Ca^{2+}$ (Fig. 6e) Of 49 neurons tested, 18 responded to ATP (10 μM), and all responses were abolished when the bath solution lacked $Ca^{2+}$, implicating P2rx receptors.

Next, we stimulated sequentially with ATP (10 μM), 5HT (10 μM), and GABA (100 μM). Representative traces of the responses are shown in Fig. 6f. Responses to GABA were measured as the change in $Ca^{2+}$ response to KCl in the absence and presence of GABA (i.e. GABA-mediated inhibition; Supplementary Fig. 8d, e). In aggregate, the large majority of neurons displayed a GABA-mediated decrease of $Ca^{2+}$, with inhibition ranging from barely detectable to 100% inhibition. The peak amplitudes of $Ca^{2+}$ responses in 148 cells to all three transmitters were analyzed by unsupervised HCA (Fig. 6g). In the dendrogram, sensitivity to ATP and 5HT drove the first- and second-order groupings respectively. Responses to GABA did not appear to track with the cell type classification, consistent with immunostaining for Gabra1 (Fig. 6d).

We assessed whether the pattern of responses to the three transmitters resembles expression of the cognate receptors in sequence data (Fig. 6a). The P2rx3 receptor is known to desensitize rapidly (<100 ms) while the P2rx2 receptor desensitizes > 100-fold more slowly[28]. Thus, we surmise that the broad responses to ATP that we recorded are derived entirely from P2rx2-expressing neurons. Neurons which express only P2rx3 are

unlikely to yield visible $Ca^{2+}$ responses to ATP that can be resolved at our imaging rate of 1 frame/1.5 s. P2ry receptors are unlikely to generate $Ca^{2+}$ responses in these experiments (Fig. 6e) because responses are dependent on extracellular $Ca^{2+}$, and ATP does not strongly activate either of the expressed P2Y receptors; P2ry14 responds principally to UDP-glucose and UDP-galactose, while P2ry1 and P2ry12 respond to ADP[29, 30]. Thus, responses to

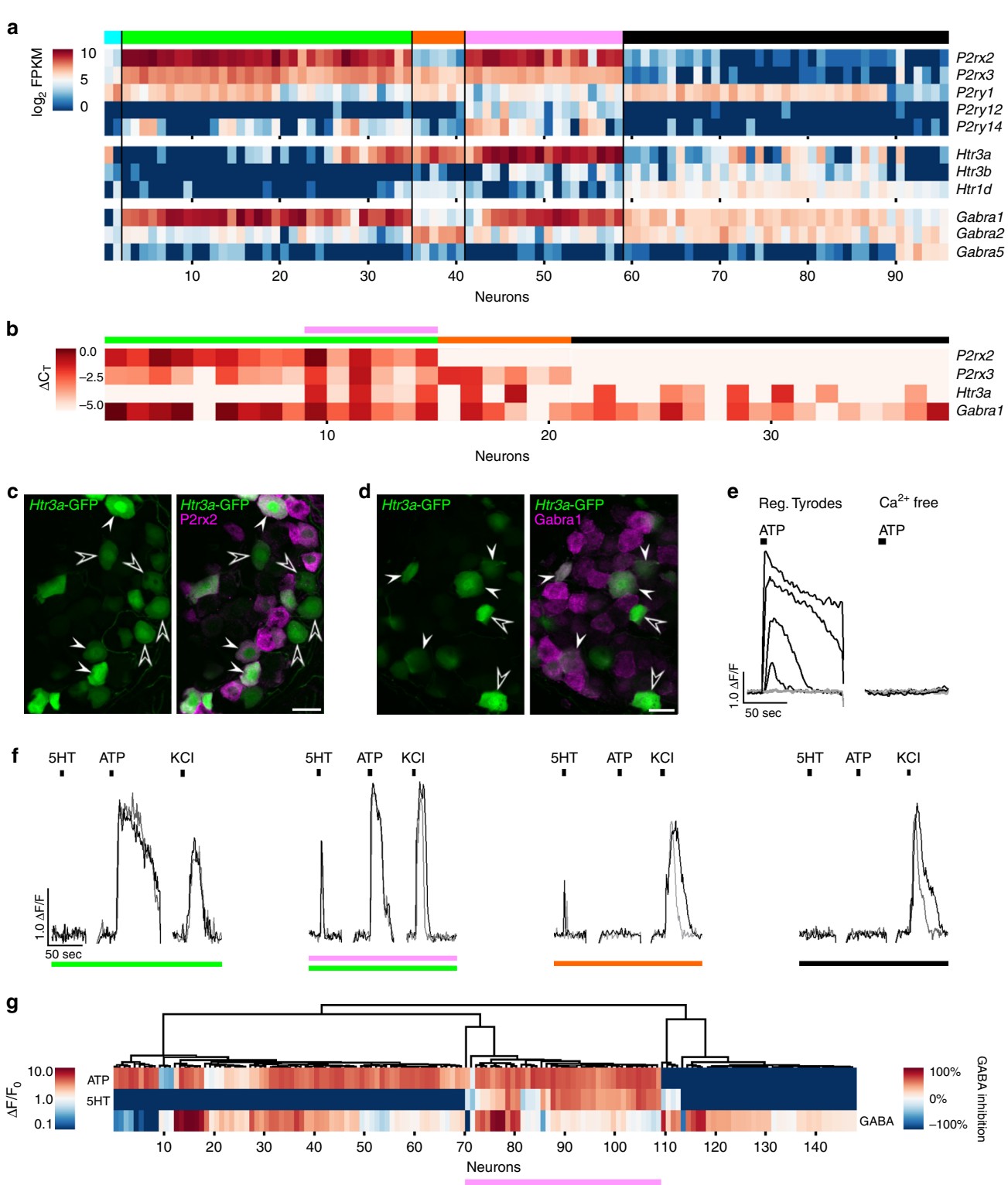

ATP and 5HT can reasonably be attributed to the expression of P2rx2 and Htr3a, respectively. The pattern of responses seen in individual traces and in the dendrogram (Fig. 6f, g) could be overlaid *post-hoc* with a classification based on gene expression. Cells which responded to ATP but not 5HT resemble T1 neurons; cells which respond to both ATP and 5HT resemble T1 or T3 neurons; and cells that responded to 5HT but not ATP resemble T2 neurons (colored bars in Fig. 6f, g).

Recent reports[26, 31] have proposed that *Htr3a*-GFP marks gustatory neurons that selectively innervate sour-sensing taste bud cells. In contrast, our transcriptome data indicate that *Htr3a* is expressed quite broadly within all sub-types of gustatory and even some somatosensory neurons (Fig. 7a). Although many of the highest FPKM values for *Htr3a* were in neurons of sub-cluster T3, more than half (49) of all the sequenced neurons expressed *Htr3a* at significant levels (FPKM ≥ 20). To confirm if geniculate ganglia from *Htr3a*-GFP mice also show this broad pattern of transgene expression, we stained sections with antibodies against GFP and cluster markers (defined above). GFP-immunoreactivity was detected in many Prrxl1-immunoreactive (i.e somatosensory) neurons (Fig. 7b), some Foxg1+ (T1) and most Mafb+ (T2) neurons (Fig. 7d), mirroring the pattern from sequence data (Fig. 7a).

We also quantified the fluorescence intensity of native GFP (i.e. without anti-GFP) in *Htr3a*-GFP geniculate ganglion neurons in cryosections that were counterstained with anti-Prrxl1 or anti-Mafb and anti-Foxg1. GFP fluorescence intensity in neurons was detected as a continuum (Fig. 7c, e). Many neurons with visible GFP fluorescence contained nuclei that were stained for Prrxl1, Mafb, or Foxg1. Thus, our interpretation is that neither *Htr3a* itself, nor GFP in this transgenic line mark a homogeneous cell type population in the geniculate ganglion. Most of the brightest GFP neurons may belong to taste sub-cluster T3, but it is not possible to distinguish these from GFP-expressing neurons of other sub-clusters by GFP intensity alone. We also note that Ca$^{2+}$ responses to 5HT were obtained in ATP-responsive and non-responsive neuron (Fig. 6f, g). That is, our functional data support the interpretation that Htr3a does not mark a homogeneous class of neurons.

**Mechanosensitivity in taste neurons**. To begin exploring the functional significance of gustatory sub-clusters, we focused on taste sub-cluster T2 (orange). In the first-level PCA (Fig. 2b), these neurons occupied a position partway between the majority of taste neurons on the left, and somatosensory neurons on the right (displayed in Fig. 8a by re-coloring the original PCA graph). In this small T2 sub-cluster, the expression of several hundreds of

genes is taste-like (Supplementary Fig. 4b). Yet importantly, many T2-selective genes (Fig. 8b) are associated with mechanosensitive neurons in trigeminal or dorsal root ganglia, including *Hs3sT2*[32], *Runx3*[33], *Gfra2*[34] and *Spp1*[35]. We also observed genes that are shared between sub-cluster T2 and the somatosensory neurons of the geniculate ganglion (e.g. *Pcdh7*, *Cadps2*). Indeed, *Mafb*, which we use as a marker of T2 neurons, is also expressed at lower levels in the somatosensory neurons (Fig. 5). Because electrophysiological recordings from taste afferents reveal responses to mechanical and other somatosensory stimuli[4, 22, 36, 37], the combination of gustatory and somatic transcriptomes in these neurons is noteworthy.

A caveat regarding the 6 neurons of sub-cluster T2 is that if one gustatory and one somatosensory neuron were captured instead of a single one, both sets of genes would appear to be expressed. To assess this possibility, we displayed the pattern of expression for ≈300 taste-selective genes on a heat map (Supplementary Fig. 4b). While neurons in sub-cluster T2 do slightly under-express taste genes relative to other taste neurons, their pattern is more similar to taste than somatosensory neurons. The numerous genes selectively expressed in sub-cluster T2 neurons (Fig. 8b) also is incompatible with dual-capture.

To confirm experimentally that neurons in sub-cluster T2 are taste neurons, we anterogradely-labeled taste neurons and immunostained ganglia for the T2 marker, Mafb (Fig. 8c). Nearly all Mafb-bright immunoreactive neurons were dextran-labeled via the CT or GSP nerves, confirming their status as gustatory neurons.

To trace the central projections of sub-cluster T2 neurons, it was necessary to identify an axonally transported protein (Mafb labels only nuclei). From the sequencing data, we identified Calbindin 1 (Calb1) as a surrogate (Fig. 8b) and confirmed, that in geniculate ganglia, Mafb+ nuclei are consistently surrounded by Calb1+ cytoplasm (Fig. 8d). Next, we double-stained coronal sections of hindbrain with Calb1 and P2rx2. The majority of central projections of taste neurons into the nucleus of the solitary tract, (NST) are labeled by P2rx2[38] as shown schematically (Fig. 8e). Our data on expression of neurotransmitter receptors (Fig. 6a, b) shows that T2 neurons express very little, if any, *P2rx2*. Correspondingly, we observed that fibers immunopositive for either P2rx2 or Calb1 (but not both) entered the lateral hindbrain together (Fig. 8f) and traverse dorsally and medially in a common bundle (Fig. 8g). However, the rostral-central subdivision of the NST, which contained nearly all of the P2rx2+ terminal field did not contain Calb1+ terminals (Fig. 8h, i). Instead, we detected a sparse Calb1+, P2rx2-negative terminal

---

**Fig. 6** Gustatory neuron sub-clusters and the somatosensory neuron cluster differ in their neurotransmitter phenotype. **a**. FPKM values for the most prominent purinergic, serotonergic, and GABAergic receptors expressed in 96 sequenced geniculate ganglion neurons. Neurons are arrayed as in Figs. 4a and 5b, with T1 (green), T2 (orange), and T3 (pink) sub-clusters of gustatory neurons followed by somatosensory (black) neurons. **b**. Heat map based on single-cell RT-qPCR in 38 manually isolated geniculate ganglion neurons. Based on expression pattern for *P2rx2*, *P2xr3*, *Htr3a*, and *Gabra1*, these neurons were classified *post-hoc* into the sequencing-derived clusters. **c**. Geniculate ganglion from *Htr3a*-GFP mouse reveals that some GFP+ neurons are P2rx2-immunoreactive (filled arrowhead) while other GFP+ neurons lack P2rx2 (Δ). GFP is visualized by native fluorescence, without immunostaining. **d**. Immunostaining for Gabra1 in a geniculate ganglion from a *Htr3a*-GFP mouse. The majority of neurons are brightly or faintly immunoreactive for Gabra1, consistent with sequencing data. Occasional neurons that are GFP-bright but lack Gabra1 (unfilled triangle) may represent gustatory sub-cluster T2 or the somatosensory cluster. **e**. Representative traces from Ca$^{2+}$ imaging of GCaMP3-expressing geniculate ganglion neurons, acutely dissociated, cultured and stimulated with ATP (10 μM) in the presence or absence of extracellular Ca$^{2+}$ in the bath. Traces for four neurons that responded (black) to ATP and three that did not (grey) are shown. **f**. Representative traces showing different patterns of responses from four neurons that were sequentially stimulated with 10 μM ATP, 10 μM 5HT and 50 mM KCl. Superimposed black and grey traces are replicate responses of the same cell. **e**. HCA of Ca$^{2+}$ responses in 148 neurons to ATP, 5HT and GABA (3 experiments, 12 ganglia). The dendrogram sorted all 148 neurons into four groups based on patterns of responses. The heat map below the dendrogram shows the magnitude of responses to neurotransmitters (ΔF/F for ATP and 5HT; % inhibition by GABA). 100% inhibition indicates complete loss of KCl response; −100% indicates a 2-fold increase in Ca$^{2+}$ response to KCl (see Methods and Supplementary Fig. 8d, e). Groupings were re-assigned with RNAseq clusters (colored bars) *post-hoc* based on the correspondence between response patterns and expression of cognate receptors. All *scale bars*, 20 μm

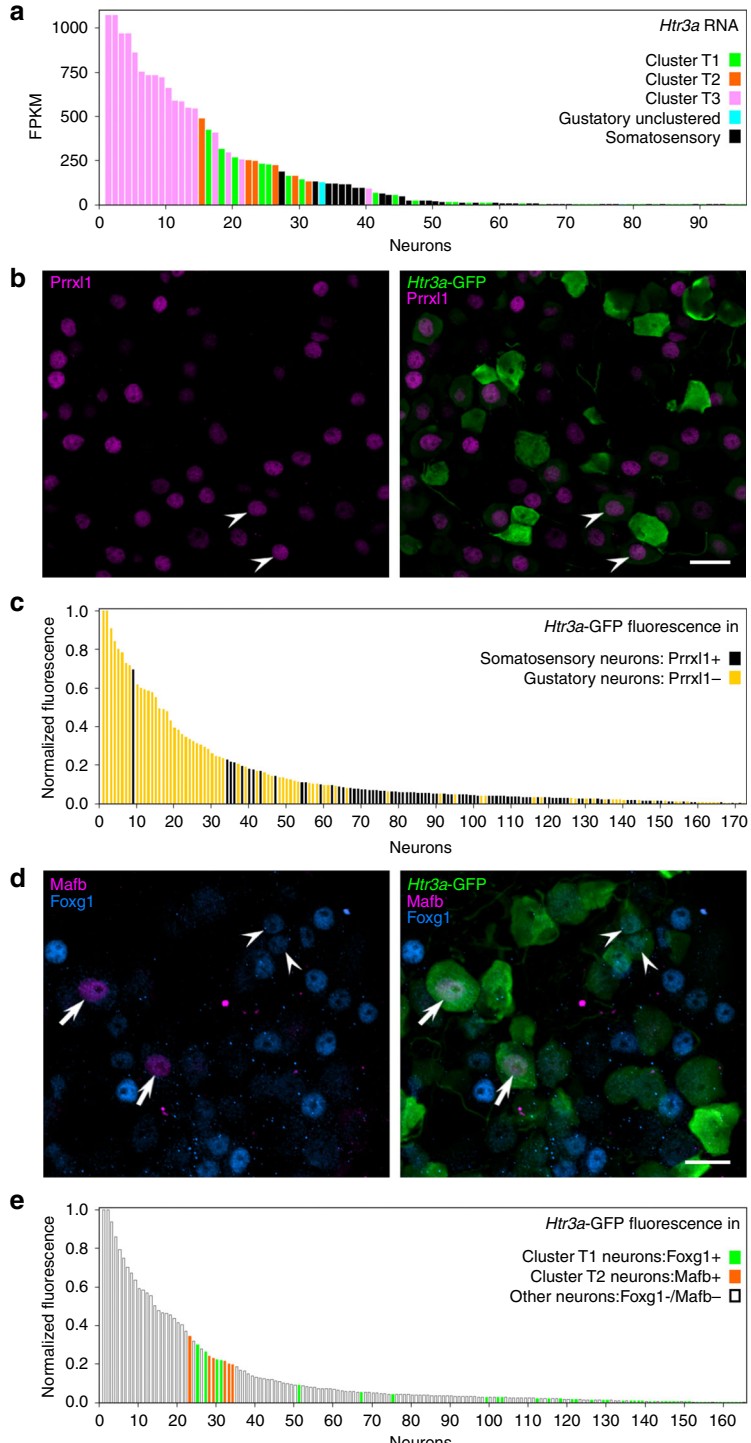

**Fig. 7** Htr3a does not mark a natural grouping of neurons. **a**. Bar graph of FPKM values for *Htr3a* in 96 sequenced neurons. Bars are colored according to the sub-cluster identity of each neuron according to HCA (Figs. 2a, 4a). **b**. Cryosection of geniculate ganglion from *Htr3a*-GFP mouse, immunostained for Prrxl1. Some somatosensory neurons in the ganglion are GFP+ (filled arrowhead). **c**. Native GFP fluorescence was quantified (using Image J) in *Htr3a*-GFP ganglion cryosections similar to **b**. GFP fluorescence intensity is a continuum and is displayed for Prrxl1+ (somatosensory, black) and Prrxl1-negative (gustatory, yellow) neurons. **d**. Cryosection of ganglion from *Htr3a*-GFP mouse, immunostained for Foxg1 and Mafb (markers for sub-clusters T1 and T2, respectively). GFP+ neurons are seen with unstained nuclei, with Foxg1-immunoreactive (arrow) or Mafb-immunoreactive (filled arrowhead) nuclei. Here, only the brightest Mafb+ (i.e. T2) neurons can be seen. **e**. Native fluorescence of GFP was quantified in all neurons in sections similar to **d**. GFP intensity in T1 and T2 neurons is displayed in colors assigned previously. Bar graphs in **a**, **c** and **e** show that Htr3a expression does not demarcate a specific group of neurons in the ganglion. *Scale bars*, 20 μm

field in the rostral-lateral sub-division of the NST, an area immediately lateral to the prominent P2rx2+ field. The relative sparseness of these terminals, compared to the P2rx2+ terminals, is consistent with the relatively small number of T2 neurons. Interestingly, the rostral-lateral subdivision of the NST is known to contain the terminal arbor of a subset of chorda tympani neurons that express P2rx3 but *not* P2rx2[38].

## Discussion

By examining the transcriptome of 96 geniculate ganglion neurons, an unbiased sampling of over 10% of the entire ganglion, we found marked distinctions across its resident gustatory and non-gustatory (somatosensory) neurons. Our study identifies several molecular markers that distinguish these cell types, and define more subtle divisions among gustatory neurons.

The geniculate, petrosal and nodose ganglia have long been known to contain both gustatory and somatosensory neurons and their relative numbers have been documented using fluorescent tracers[39]. Previous efforts to distinguish somatosensory from gustatory neurons in the geniculate ganglion have been principally in the context of embryonic fate specification and targeting. The geniculate ganglion is believed to develop as the fusion of two ganglia that have separate placodal origins[17, 40]. *Phox2b* was shown to be an essential switch for the differentiation of embryonic neurons of geniculate, petrosal and nodose cranial ganglia (VII, IX, X nerves) into visceral sensory neurons[17]; in *Phox2b*-null mice, taste and other visceral sensory neurons develop central projections that resemble those of somatosensory neurons. *Pou4f1*, *Pou4f2* and *Prrxl1* are widely acknowledged to represent the molecular signature of trigeminal somatosensory neurons[17, 40], consistent with their identification here as markers for non-gustatory neurons of the geniculate ganglion. In addition, *Hmx1*, a homeo-domain transcription factor, was studied during embryonic development of somatosensory neurons in several cranial ganglia including the geniculate[41]. *Hmx1* is involved in facial morphogenesis rather than neurogenesis or specification of a sensory modality. In our data set from adult ganglia, *Hmx1* is detected only at modest levels, and only in a small subset of somatosensory neurons (#90–96 in Fig. 5b). Other transcription factors, including *Six1*, *Eya1* and *Eya2* (Fig. 5b) also have previously been associated with embryonic development of geniculate and petrosal ganglia, both of which contain taste neurons (reviewed,[42]).

The neurotransmitter phenotypes of geniculate ganglion neurons are heterogeneous. Our $Ca^{2+}$ imaging data suggest a clear distinction between neurons that respond to ATP and those that do not (Fig. 6g). This is consistent with our finding that all gustatory neurons express *P2rx2* and/or *P2rx3*, whereas somatosensory neurons in this ganglion (which innervate the pinna) express very low levels, if any, of *P2rx2*, *P2rx3* (Fig. 6a). The metabotropic P2ry1, which is expressed in the somatosensory neurons, is insensitive to ATP, but instead, is activated by ADP which often contaminates ATP solutions[29, 30]. The ionotropic serotonin receptors *Htr3a* and *Htr3b* are broadly expressed across the geniculate ganglion without apparent relation to neuronal cell types. This does not support the interpretation that there is a unique class of geniculate ganglion neurons that express Htr3a[26, 31]. Expression levels of *Htr3a* are highly variable, and the functional significance of 5-HT signaling for all the gustatory and somatosensory cell types that express it remains unresolved. Finally, we observed a wide range of responses to GABA. This may reflect variable expression levels of GABAA receptors receptors and/or $Cl^-$ gradients. No correspondence of GABA responses with neuronal cell types was apparent in our data.

The significance of the taste sub-clusters, T1, T2 and T3, revealed in the present analyses remains to be elucidated. One possibility is that the taste subclusters represent contributions from afferent fibers in the two distinct gustatory nerves (CT, GSP) and the different receptive fields they innervate (fungiform taste buds of anterior tongue, foliate taste buds of lateral tongue, and taste buds of palate and naso-incisor duct). Indeed, neurons of these sub-clusters might influence or direct the morphological and molecular differences that have been noted in taste buds from these different regions[43]. A second possibility is that the sub-clusters represent different neuronal states[12, 13, 16]. For instance, gustatory neurons exhibit plasticity that is dependent on nutritional state[44] or on synaptic interactions with peripheral and central targets[45]. However, the sub-clusters are deeply separated in the dendrogram (Fig. 4a), with dozens of genes contributing to the principal components (Figs. 4b, 5a). Moreover, they display distinct neurotransmitter phenotypes (Fig. 6). All these factors, along with the absence of neurons displaying intermediate transcriptomes, suggest that T1, T2, and T3 represent different cell types, not different states. A third possibility is that the sub-clusters represent neurons that serve different physiological functions. For example, parallel pathways have been proposed for taste perception vs. physiological reflexes or for circuits driving ingestive behaviors[46]. There is also the possibility that the gustatory sub-clusters are composed of neurons transmitting distinct taste qualities. That is, each sub-cluster could be dedicated to sweet or bitter or sour etc. Although geniculate ganglion neurons responding only to only a single quality (taste "specialists") have been reported[3, 5, 6, 47], it has never been shown that such restricted tuning is intrinsic to the neurons. To the contrary, individual neurons change their tuning at different stimulus concentration[6]. In sum, there is limited evidence directly supporting any of the above possibilities, which, in any case are not mutually exclusive. The marker genes that our study reveals will permit direct tests of whether distinct neuron types innervate taste bud cells that express different taste receptors. This is an explicit prediction of "labeled line" coding for taste[48].

The largest taste subcluster, T1 (green), shows evidence of additional subdivisions (see dendrogram and correlogram, Fig. 4a). While we have not yet explored these subdivisions in detail, an example can be seen by comparing heat maps for transcription factors and neurotransmitter receptors (Figs. 5b and 6a respectively). Neurons #27–35 of subcluster T1 in each heat map express the combination of transcription factor *Eya2* and 5HT receptor *Htr3a*. The same set of 9 neurons stands out as a separate grouping within T1 when evaluated across 580 taste-selective genes (Fig. 4a). Somatosensory neurons #91–96 also appear to form a sub-cluster in these heat maps. More detailed analyses will be necessary to identify intersectional markers that define such sub-types and to understand their significance.

Taste sub-cluster T2 (orange) reveals one possible functional significance. Many of the genes selectively expressed in T2 neurons have previously been associated with mechanically sensitive somatosensory neurons, including *Hs3sT2*[32], *Runx3*[33], *Gfra2*[34], *Cdh1* and *Spp1*[35]. Recordings from single chorda tympani afferent fibers[49] and from geniculate ganglion neurons[37] reveal cells that respond to mechanical and other general sensory stimuli[4, 50, 51]. Responses to oral mechanical sensation were recently recorded in central neurons in an area of the NST that is slightly displaced relative to chemical sensation[52]. These responses may represent trigeminal somatosensory inputs as the authors suggested[52], but may also include the gustatory mechanosensory input that we postulate here. Synapses of chorda tympani neurons in the NST differ between the rostral-central vs. the rostral-lateral subdivisions. Based on their larger size and other ultrastructural features, the rostral-lateral terminals were

proposed to represent mechanosensation, while taste inputs were mostly in the rostral-central subdivision[53]. Our evidence, albeit incomplete, is that the putative mechanosensory T2 neurons of the geniculate ganglion also terminate in the same rostral-lateral subdivision and may represent at least part of the non-chemosensory input into the gustatory portion of the NST. Future studies using markers proposed here should help to identify the synaptic terminals of mechano-gustatory neurons

and the associated central pathways that they recruit. Recently, Semaphorins 3A and 7A on bitter- or sweet-sensing taste bud cells were suggested to specify synaptic contact by interacting with cognate receptors expressed on specific gustatory afferent neurons[54]. Our sequencing data indicate that two of the Semaphorin receptors proposed in the study (Nrp1, Plxnc1) are expressed primarily in somatosensory neurons that innervate the ear, not in gustatory neurons. The other three Semaphorin

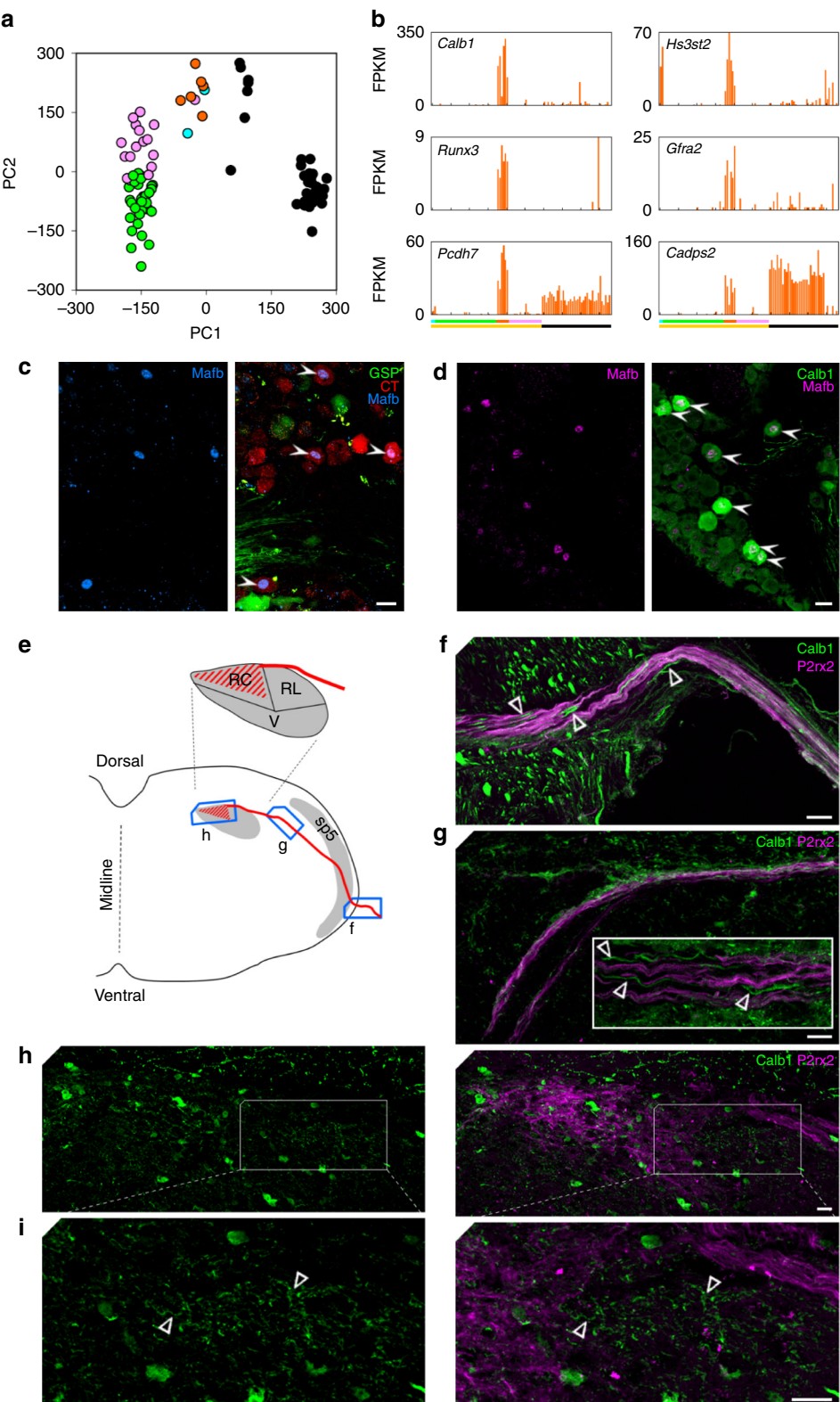

receptors proposed (Plxna4, Plxna3, Itgb1) are broadly expressed in most or all neurons of the geniculate ganglion (somatosensory and gustatory alike) and thus also are unlikely to selectively determine nerve-target interactions in taste buds.

The availability of selective markers for geniculate ganglion neuronal cell types will support more detailed neuroanatomical, physiological and developmental studies to elucidate the adequate stimuli for gustatory neurons, their functional roles and their patterns of innervation of peripheral targets and central projections.

## Methods

**Mice.** All procedures with mice were conducted according to the NIH Guide and were approved by the Institutional Animal Care and Use Committee of the University of Miami. For tissue collection, mice were killed by $CO_2$ asphyxiation followed by cervical dislocation. Alternatively, for perfusion fixation for immunohistochemical staining, mice were deeply anesthetized with a ketamine / xylazine cocktail injected i.p., $\approx 20$ ml PBS was perfused via an intracardiac cannula, followed by 40 ml of ice-cold 4% paraformaldehyde. Geniculate ganglia and brain regions were dissected, postfixed for 1 h or overnight respectively, cryoprotected in 30% sucrose overnight, and embedded in OCT for cryosectioning.

C57BL/6 J mice of both sexes were used for all procedures except where other strains are indicated.

*Pirt*-GCaMP3 mice[55] in which sensory neurons, including all those in the geniculate ganglion[6] express a genetically encoded $Ca^{2+}$ indicator, were used for the $Ca^{2+}$ imaging studies.

B6.Cg-Tg(*Gfap*-cre)73.12Mvs/J (Jackson Lab Stock # 012886): Fixed tissues from *GFAP;TdT* double heterozygotes[56, 57] were kindly provided by Michael Sofroniew, University of California, Los Angeles and were used to confirm expression of GFAP in satellite glia to measure glial contamination in dissociated ganglion preparations prior to cell capture and sequencing (Supplementary Fig. 1c, d).

*Trhr*-GFP mice (GENSAT, MMRRC stock # 030036-UCD): Breeding mice were kindly provided by Marla Feller, University of California, Berkeley and have been previously validated[58]. We further show (Supplementary Fig. 5d, e) that expression of GFP follows the pattern expected from *in situ* hybridization of the endogenous *Trhr* gene.

*Htr3a*-GFP mice (GENSAT, Tg(Htr3a-EGFP)DH30Gsat): GFP is expressed in cells that endogenously express a principal ionotropic receptor for serotonin. The pattern of transgene expression was previously validated by overlap with endogenous gene products by in situ hybridization and immunostaining[27, 59]. Breeder mice were kindly provided by Chris McBain, National Institutes of Health.

**Ganglion dissociation and enrichment for neurons.** Geniculate ganglia ($\approx 900$ neurons) were rapidly dissected from six 8–12 week-old C57BL/6 J mice of both sexes and placed in Ca-Mg-Free Hank's Balanced Salt Solution (HBSS, Thermo-Fisher 14170161) at room temperature. All 12 ganglia were sequentially digested as described[60], first with 1 ml of 40 u/ml Papain with NaHCO3 and l-cysteine at 37 °C for 13 min, followed by 4 mg/ml Collagenase Type 2, 4.7 mg/ml Dispase Type 2 at 37 °C for another 13 min. Enzymes were Papain (Worthington, Lakewood, NJ, #LS003126, Collagenase (Worthington, # LS004174) and Dispase (Roche 04942078001). Ganglia were washed with Tyrode's buffer (in mM: 135 NaCl, 5 KCl, 2 CaCl2, 1 MgCl2, 10 HEPES, 5 NaHCO3, 10 glucose, 10 pyruvate), adjusted to pH 7.4 (318–323 mOsM) to remove enzymes and transferred into 40 µl fresh Tyrode's solution. Under visual guidance under a stereomicroscope, ganglia were gently triturated 3–5 times each with a 200 µm diameter and then a 140 µm fire-polished glass pipet. Large debris was removed by gravity filtering the suspension through a $\approx 35$ µm cell sieve (Corning 352235) into 1 ml Tyrode's buffer in a 15mm

cell culture well. At this stage, the suspension still included significant quantities of axon fragments and glial cells (Supplementary Fig. 1a). The cell suspension was manually swirled such that neurons remained concentrated at the center of the well while glial cells and axon fragments dispersed centrifugally to the perimeter from where they could be withdrawn in 700 µl aliquots, aspirated from the perimeter. By repeating this manual separation 8–10 times with 700 µl of fresh Tyrode's buffer added and withdrawn at each iteration, the suspension became enriched for neurons and substantially depleted of satellite glia and axon fragments (Supplementary Fig. 1b). Total time for ganglion dissociation and neuronal enrichment was $\approx 60$ min.

These procedures were optimized for neuronal enrichment, yield and neuronal viability by monitoring propidium iodide exclusion during several preliminary trials. We also monitored depletion of glial cells by RT-qPCR for Glial Fibrillary Acidic Protein (*Gfap*) which is expressed in geniculate ganglion satellite cells and Schwann cells in the afferent nerve (Supplementary Fig. 1c). While *Gfap* mRNA was readily detected in the intact ganglion, the purified neuron preparation showed $\geq 10^5$-fold depletion of *Gfap* relative to *Snap25* mRNA, as assessed by the $\Delta\Delta Cq$ method (Supplementary Fig. 1d).

**Cell capture, cDNA synthesis, pre-amplification.** Two trial preparations of geniculate ganglion neurons were used to optimize the buoyancy of the suspension medium to match that of the neurons, with reagents and protocol from Fluidigm. Cell viability was assessed in an aliquot of each final preparation by trypan blue exclusion on a Countess Automated Cell Counter (ThermoFisher). Two independent cell suspensions (on separate days) that displayed neuronal viability of 93 and 96%, respectively and viability of glia ($\leq 12$µm diameter) at 50%$\leq$ were used for subsequent steps. The suspension included $\approx 350$ cells/µl in a neuron: glia ratio $\geq 1.5$. Each cell suspension was loaded onto a primed Fluidigm C1 System (located in the University of Miami CFAR Core). Capture was on a Fluidigm integrated fluidics circuits (IFC) for large cells ("17–25 µm", Fluidigm 100–5761) exactly according to manufacturer's instructions. After completion of the run, all capture sites on the chip were examined microscopically and photographed. From a total of 192 sites on two IFCs, we rejected 68 that contained more than one cell or in which captured neurons showed blebs or membrane ruffling (Supplementary Fig. 2d, e). Captured neurons ranged from 19–40 µm in diameter (see examples, Supplementary Fig. 2a–c).

The C1 instrument was then primed with reagents for automated cell lysis, cDNA synthesis and pre-amplification. Cells were lysed in the presence of Control RNA Spikes (ThermoFisher AM1780). Full-length cDNAs were synthesized and pre-amplified for a standard 22 cycles using the SMART-Seq v4 Ultra-Low Input RNA Kit for the C1 system (Clontech) according to the Fluidigm manual. Within 20 min of script completion, cDNAs were harvested from the instrument, diluted into a 96-well plate, and stored at −80°. Approximately 7% of each sample was removed for first-level quality controls including quantification and full-length assessment on Agilent 2100 Bioanalyzer. Yields ranged from 13 to 28 ng (mean 21 ng) cDNA per cell. cDNAs displayed average lengths of 1660 bp (median 1860 bp).

An aliquot (0.4%) of cDNA from every singly captured cell was tested by qPCR for *Snap25* as well as *P2rx2* and/or *Cacna1a*, representing abundant, moderate and relatively rare mRNAs (Supplementary Fig. 2f). We selected 48 cells from each of two completed runs on the C1 system, based on low Cq for *Snap25* (indicative of best RNA/cDNA yield). The pre-amplified cDNA for those 96 cells was transferred to a fresh 96-well plate for sequencing.

**Sequencing and data analyses.** Synthesis of cDNA libraries and sequencing were performed at the Next-Generation Sequencing Core (NGSC) at the University of Pennsylvania. An Illumina Nextera XT DNA library preparation protocol, modified for single-cell mRNA sequencing of cDNAs acquired on the C1 system was used. Each single cell library was barcoded during synthesis, libraries were pooled (median length before fragmentation was 900 bp) and single-end sequenced for a total depth of $250 \times 10^6$ reads for the pool, running on a single lane on an Illumina

**Fig. 8** Mechanosensory-like taste neurons of the geniculate ganglion. **a**. PCA from Fig. 2b, re-colored as assigned in Fig. 4a to display taste sub-clusters, T1, T2, T3, and somatosensory cluster. Neurons of sub-cluster T2 (orange) appear midway between other taste neurons and the somatosensory (black) neurons of the ganglion. **b**. Expression in all 96 neurons of 4 genes (Calb1, Hs3sT2, Runx3 and Gfra2) that are selectively expressed in T2 neurons, and 2 genes (Pcdh7, Cadps2) that are shared between T2 neurons and somatosensory neurons of the ganglion. Cluster and sub-cluster assignments are indicated below the bar graphs. **c**. In spite of expressing some somatosensory genes, T2 neurons are *bona fide* taste neurons as shown by the presence of anterograde labeling dye in the cytoplasm surrounding each Mafb+ nucleus (25 of 27 Mafb+ nuclei counted in 9 sections from 4 mice; filled arrowhead). **d**. Neurons of sub-cluster T2 (i.e. Mafb+, filled arrowhead) selectively express calbindin 1; few other neurons in the ganglion express this cytosolic protein. We detected this overlapping expression in 21 of 22 Mafb+ neurons in cryosections from 3 mice. **e**. Schematic of coronal section through hindbrain showing the trajectory of taste fibers in the hindbrain and terminating in the Nucleus of the Solitary Tract (NST, grey inset). Regions within the NST include the rostral-central (RC), rostral-lateral (RL) and ventral (V). Boxes depict the positions of micrographs **f–h**. Calb1+ fibers (unfilled triangle) enter the hindbrain (**f**) and traverse dorso-medially (**g**) alongside other taste afferent fibers, most of which are P2rx2-immunoreactive. The T2 fibers are Calb1+ but P2rx2-negative. **h**. The P2rx2+ neurons produce a triangular terminal field dorsomedially within the NST. The boxed area is shown at higher magnification in **i**. Terminal arborization of the sparse Calb1+ fibers (unfilled triangle) is most visible immediately lateral to the P2rx2+ terminal field. *Scale bar*, 20 µm

HiSeq 4000 100SR system with HiSeq Control Software 2.2.58. Base-calling was performed using RTA: 1.18.64. Data were demultiplexed and BCL files converted to FASTQ using bcl2fastq 2.17.1.14. Sequence data were processed at the NGSC through their RUM package[61] v2.0.5_06 for genome alignment, junction calling, and feature quantification. Reads were mapped against mouse transcriptome (mm9) using Bowtie and against genome using Bowtie and Blat, and then converted to mapped Fragments Per Kilobase of exon per Million reads (FPKM).

Sequence data were analyzed in Python (v 2.7.12). Diagrams were created using the python modules Matplotlib (v 1.5.1)[62] and Seaborn (v 0.7.1) (https://seaborn.pydata.org/). Principal component analyses (PCA) were performed using the PCA module from Python Sci-kit Learn (v 0.18.1)[63]. Management of dataframes in Python was performed using the Python modules Pandas (v 0.17.1)[64] and NumPy (v 1.11.0)[65].

We performed unsupervised hierarchical clustering of cell-to-cell Pearson correlation coefficients (PCCs) and Principal Component Analysis. Dendrogram linkage was determined via the Unweighted Pair Group Method with Arithmetic Mean (UPGMA). Distances in the dendrograms represent centered Pearson correlation coefficients.

The first-level hierarchical clustering (HCA) was performed on all 96 cells. We reasoned that many low- and moderately-expressed genes may be shared across all geniculate ganglion sensory neurons while major groupings of neurons may be differentiated on the basis of highly expressed genes. Thus, for this initial HCA, we filtered for genes with average expression level of 200 FPKM or higher across all 96 cells (333 genes, Fig. 2a). First-order of separation in the dendrogram was used to assign group IDs (yellow or black) which were retained for each cell before running the PCA on all 96 neurons using all detected genes ($n = 17,225$ genes, Fig. 2b). Although as many as four subclusters can be distinguished in the HCA and PCA of all 96 neurons, a priori knowledge that the geniculate ganglion consists of two anatomically and functionally distinct sets of neurons (gustatory and somatosensory) led us to focus the initial clustering on the first-order of separation. Parenthetically, we also ran the HCA on genes filtered for a much less selective criterion, of FPKM > 10 in at least 3 neurons (yielding 8748 genes). We obtained the identical division of 96 neurons into gustatory (yellow) and somatosensory (black) clusters.

The larger group ($n = 59$ neurons, yellow) underwent second-level hierarchical clustering. Pearson Correlation Coefficients (PCCs) were calculated for genes that met the following criteria: (i) expression in at least 2 neurons, (ii) average FPKM ≥ 1 in taste neurons, (iii) ratio of FPKM ≥ 5 in taste relative to somatosensory neurons. The first two criteria decrease technical and biological noise by eliminating genes from consideration that are (i) highly expressed in just one cell or (ii) are poorly expressed overall. The third criterion focuses this second-level HCA on taste-selective genes, which are the most likely to underlie taste neuron subtypes. These filtering criteria yielded clustering of neurons into 3 sub-groups, which were assigned labels T1, T2 and T3. The group affiliation of 2 of the 59 neurons could not be determined. Parenthetically, we obtained identical taste sub-clusters when the third criterion (above) was set to ≥2-fold (1327 genes), ≥5-fold (580 genes) or ≥10-fold (341 genes). The consistency of the HCA outcomes suggests that the classification has a biological rather than a technical basis.

We also performed a second-level PCA on the 59 taste neurons based on a single criterion of FPKM ≥ 10 in at least 3 taste neurons. Here, the filter was selected only to reduce noise in the analysis by eliminating poorly expressed genes while retaining the maximum possible number of genes. Data from 8150 such genes were incorporated into this alternative second-order PCA and yielded the same groupings as the second-order HCA.

Next, we performed PCA on the 59 gustatory neurons using a list of genes encoding transcription factors and co-factors and chromatin remodeling factors (collectively, TF). We merged lists from the Riken TF (http://genome.gsc.riken.jp/TFdb/tf_list.html) and Bioguo TF (http://bioguo.org/AnimalTFDB/species.php?spe=Mus_musculus) databases and then filtered the combined list for those TFs expressed at average FPKM ≥ 1 across all gustatory neurons. As described above, this criterion simply removes noise by limiting the analysis to genes that are actually expressed. 976 of 2407 of the TF genes met this criterion.

**Single-cell and bulk RT-PCR.** Tissues and single cells were processed for RNA extraction, cDNA synthesis and PCR exactly as previously reported[23, 66]. In brief, dissociated neurons (Supplementary Fig. 1a) were manually isolated using fire-polished 24 μm diameter glass pipets from a suspension of dissociated ganglia.

For end-point RT-PCR, geniculate ganglia or individual neurons were lysed in the presence of 200 ng poly-I (Sigma P4154) and RNA was extracted using Absolutely RNA Nanoprep kit (Agilent 400753). For RT-qPCR, single neurons were lysed (Clontech 635013) and mRNAs were directly converted to cDNA without purification. Superscript III (ThermoFisher #18080–044) was used for all reverse transcription. 2.5–12% of each cell cDNA was used as template in PCRs. End-point PCRs for single cells were limited to 40 cycles (except for transcription factor genes which required 45 cycles). The concentration for each mRNA in each sample was normalized to Snap25 mRNA concentration in the same sample using the ΔΔCt method. SYBR Green Master Mix (Bio-Rad # 1725264) on a Bio-Rad CFX system was used for qPCR. A list of PCR primers and corresponding annealing temperatures is in Supplementary Table 1.

**Antibody staining and imaging.** All antibodies used are listed in Supplementary Table 2, along with how they were validated. We validated anti-Phox2b in-house (Supplementary Fig. 5a, b). Cryosections of ganglia (20 μm thick) or brain regions (40 μm thick) were processed as previously detailed[67]. Briefly, sections were permeabilized in 1% triton in PBS, blocked in 5–10% normal donkey serum for 2 h at room temperature, and incubated at room temperature with primary antibodies diluted in 5% donkey serum. Diluted antibodies against P2rx2 and Gabra1 were incubated for 3 h while all other antibodies were left to bind overnight. After washing off the primary antibody, sections were incubated in secondary antibodies conjugated with fluorescent Alexa dyes, diluted in 5% donkey serum for 1 h before final washing and mounting.

Fluorescent micrographs were captured on a Fluoview FV1000 laser scanning confocal microscope using Fluoview FV10-ASW-UP4 v1.7 software. All images were captured within the linear range of output.

Immunoreactivity was quantified from unprocessed images using Fiji (ImageJ 1.51k). Regions of Interest (ROIs) of one size were set for all neurons. Background was set as the average of the four weakest neuronal ROIs of each image. Values are presented as background-subtracted fluorescence, normalized to maximum fluorescence in the same image.

**Anterograde labeling.** The chorda tympani and greater superficial petrosal nerves (gustatory branches of cranial nVII) were exposed in anesthetized mice and cut immediately distal to the tympanic bulla. Crystals of fluorescent dextran were placed on the cut proximal end as previously described[21]. Dyes selected were lysine fixable 3 kDa dextrans, conjugated to fluorescein or tetramethylrhodamine (ThermoFisher D3306, D3308). After 5–7 h to allow dextran to accumulate in the neuronal soma, mice were fixed by perfusion and ganglia were harvested and processed for immunohistochemical analyses as described above.

**Ca²⁺ imaging.** A suspension of neurons from dissociated geniculate ganglia of Pirt-GCaMP3 mice (3–4 months old) was plated onto Cell-Tak-coated glass coverslips and incubated in F12 medium (ThermoFisher 11765–054) with 10% fetal bovine serum for ~ 2 h at 37° in a $CO_2$ incubator. This incubation permits neurons to recover from axotomy and enzymatic digestion[60]. Parenthetically, this culture step promoted the health of neurons such that over 70% of the cells that attached to the substrate yielded robust and reproducible responses to KCl (Supplementary Fig. 8a, b,c). Coverslips were transferred to a recording chamber (Warner Instruments) filled with Tyrode's buffer and perfused at 2 ml/min at room temperature. Imaging was carried out on an Olympus FV1000 laser scanning confocal microscope with a 20x water immersion objective, using Fluoview FV10-ASW-UP4 v1.7 software and a scan rate of 0.67 Hz. All stimuli were from Sigma: ATP (A6419), 5HT (H9523), GABA (A5835) and were used at 10, 10 and 100 μM respectively. These concentrations were selected as they elicit near-maximal responses from P2rx2 and P2rx3[28], Htr3a,b[68] and Gabra[69] receptors. Stimuli were bath applied for 9 s followed by at least 80 s of rinse between trials.

Scanned images were stabilized using ImageJ2, regions of interest (ROIs) were drawn around all neurons and pixel intensity data were exported to Excel. Responses (ΔF/F) were calculated as peak change in fluorescence during 40 s post-stimulus divided by the mean fluorescence of 10 frames just prior to the stimulus (i.e. baseline, F). Cells were considered as healthy and were included in the analyses if they met two criteria: (1) responses to 30 or 50 mM KCl were greater than 5 standard deviations above baseline; (2) a flat baseline preceded each stimulus and was re-established in ≤ 3 min after each stimulus.

Responses to ATP and 5HT (ΔF/F) were logged as positive if peak amplitude > 5 s.d. above baseline. We quantified responses to GABA as inhibition of KCl-evoked $Ca^{2+}$ responses relative to the mean of two flanking responses to KCl alone[70] (Supplementary Fig. 8d). The following 5 stimuli were applied in sequence: 5HT, ATP, KCl, KCl + GABA, KCl. The entire sequence was then repeated for a total of ten stimulations.

For the hierarchical clustering (Fig. 6f), we included only neurons whose responses (or lack thereof) were repeated twice for each of the three neurotransmitters. The mean of the two responses to each transmitter was tabulated for each neuron and formed the basis for hierarchical clustering. Dendrogram branches represent Euclidean distances and linkage was determined by unweighted pair-group method with arithmetic mean (UPGMA).

**Data availability.** Sequence data that support the findings of this study are deposited in GEO with the accession code GSE102443.

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

## Acknowledgements

We acknowledge the invaluable help of several colleagues for sharing resources, specifically Marla B. Feller, University of California, Berkeley for living *Trhr*-GFP mice; Chris J. McBain, Eunice Kennedy Shriver National Institute of Child Health and Human Development for living *Htr3a*-GFP mice; Michael Sofroniew, University of California, Los Angeles, for fixed tissue from *Gfap*-Cre;tdTom mice; Deolinda Lima and Carlos Reguenga, Faculty of Medicine of the University of Porto for a validated antibody against *Prrxl1*. We thank James Eberwine, University of Pennsylvania, for helpful discussions in the early phase of this project; Chengsan Sun, University of Virginia, for anterograde labeling used in preliminary experiments, Kyle Palmer, Opertech Bio, for discussions on P2 receptor pharmacology, the University of Miami CFAR Core and Oncogenomics Core for cell capture and initial quantifications and the University of Pennsylvania Next-Generation Sequencing Core for sequencing. This work was supported by grants from the National Institutes of Health to N.C. and S.D.R. (R01 DC014420), to N.C. (R01 DC006308) and to D.L.H. (R01 DC00407).

## Author contributions

Dissociation, validations and capture of ganglion neurons and cDNA synthesis was developed and executed by G.D. and N.C. Bioinformatics analyses on single cell sequencing data was by D.H., G.D. and N.C. Ca²⁺ imaging and data analyses were conducted by J.K.R., S.D.R. and N.C. Immunohistochemical analyses and confocal imaging were by D.H. and G.D. D.L.H. trained G.D. for nerve labeling and assessed all hindbrain neuroanatomy. All authors drafted and edited the manuscript.

## Additional information

**Competing interests:** The authors declare that there are no competing financial interests.

