## [Peer Review File · Nature Communications]

Reviewers' comments:

Reviewer #1 (Remarks to the Author):

This paper uses single-cell RNA-seq to analyze neurons in the mouse geniculate ganglion, focusing on gustatory system neurons. The authors combine transcriptomics, protein expression, pharmacology, and projection labeling to examine differences among the putative transcriptomic types identified through RNA-seq. The multimodal approach to dissecting these neurons is excellent, and provides a comprehensive analysis of the main subclasses of gustatory neurons in this ganglion.

Concerns:

1. Whereas the results of the hierarchical clustering are clear, the separation of clusters in principal component space is not. If the PCA plots in Figure 4b were not colored by HCA type, it would be challenging to say the sub-clusters are distinct, especially in the absence of any metrics related to cluster tightness/similarity in PC space. This is likely due to the inclusion of all genes for the PCA, as opposed to the restricted gene set for HCA - the addition of non-informative genes to the PCA is likely blurring the distinctions among these cell types. In order to maintain truly robust, independent confirmation of these clusters, this reviewer suggests using a more representative list of genes for the PCA. This can be done either by selecting genes with more variance than technical noise (if spike-ins were included in the experiment), or using the approximation that technical noise tends to follow a Poisson distribution (variance proportional to the mean) for most genes. Filtering out "noise" genes for the PCA should provide better separation, similar to that observed in the hierarchical clustering, and thus serving as independent confirmation of the clusters.
2. There is insufficient justification for the gene selection thresholds for cell type identification: the authors should specify why certain thresholds for FPKM/fold-change were chosen for gene selection. Did these give optimal separation of clusters? Did certain thresholds correspond to known degrees of gene expression separation of major classes?
3. Within the t1 subcluster, there appears to be a subcluster that is Htr3a+ as well as Eya2+. Even if there is not enough statistical power to separate this set of cells (as mentioned in the text) it would be worthwhile to comment specifically on this aspect of heterogeneity.
4. The projection labeling identifying the distinct axonal projections arising from t2 cells is very intriguing. In the discussion, the authors mention that there is a cellular population that projects to a different portion of the nucleus of the solitary tract - have the authors found where the t2 cell terminals arborize? This would be a valuable piece of information, and it leaves the reader somewhat unsatisfied to not have this information, especially in light of the discussion.

Minor concerns:

1. Supplemental Figure 5c is better served in the main text, to show co-localization of Mafb and Phox2b
2. Figure 6c would be well served with quantification of "faint" and "bright" expression - it is not very easy to tell from looking at the images.
3. On page 9, in the final paragraph, the authors specify that the desensitization for P2rx3 is rapid (~100ms); to put this in context, the rate of desensitization for P2rx2 (>100-fold slower), and the imaging rate (0.67 Hz) should be mentioned. The dramatic difference is essential to assigning the Ca response to P2rx2.
4. The legend in Figure 7a is missing the cyan element.
5. On page 10, line 21, consider adding "by GFP intensity alone" to the sentence ending in "it is not possible to distinguish these from GFP-expression neurons of other clusters"

6. On page 11, line 11, there is an extra "neurons" in the line "neurons in sub-cluster t2 neurons are taste neurons".

Overall, this is an excellent characterization of this subclass of gustatory system neurons, well-served by a multimodal approach and extensive validation of the single-cell RNA-seq data using qPCR and antibody staining, as well as validation of the clusters using projection labeling and pharmacology.

Reviewer #2 (Remarks to the Author):

This manuscript is the first report of RNAseq expression data for the neurons of the geniculate ganglion. Neurons within this ganglion have been studied for decades for their electrophysiological responses to taste stimuli. These studies have failed to culminate in any conclusion concerning how these neurons code gustatory information. One possible explanation is that these neurons exist as types. Each type may contribute to different parallel pathways and code different types of information, such that examining pooled functional data yields limited information about how information in this system is coded. Therefore the field of taste requires a basic understanding and definitions of gustatory neuron types. This manuscript provides basic genome wide expression data which represents the first step in gaining this information. The statistics used to analyze these data were appropriate and the data are convincing. These data will likely be the foundation for numerous future studies examining function, connectivity, and development of geniculate and gustatory neurons. Therefore, these data are very high impact for the field of taste. However, I do have multiple suggestions for how this manuscript might be improved.

Major Concerns:

1) NIH public access requirements dictate that the raw expression data from this experiment is to be provided online upon publication. There is no indication in the manuscript of where this link might be located in the future publication. In addition, in the supplemental reporting check list, the section on future data availability was not completed by the authors. While it is expected that these data would not be provided in peer review, some indication from the authors that that these data will be provided upon publication would have been useful for evaluation. This manuscript does not have the same high impact if these data are not provided.

2) The description of somatosensory vs gustatory innervation of the NTS was vague and the results of this experiment were unclear. Chorda tympani fibers have a dense projection into the rostrocentral subdivision of the rostral NTS and a sparse projection into the rostromedial subdivision (Corson et al., 2012). The rostromedial subdivision also receives somatosensory input from the trigeminal and is not labeled with P2X2 (Breza and Travers, 2016). It is therefore reasonable for the authors to hypothesize that if the t2 subpopulation is somatosensory in nature, it would project specifically to the rostromedial region. However, Figure 8e does not show this. While a clear P2X3 terminal field can be seen, the Calb1-staining looks like cell bodies and is distributed throughout the P2x3 field. Furthermore, it is unclear where this section is taken from rostral to caudal in the NTS or where the rostrocentral or rostromedial subdivisions are located in this image. This issue can likely be easily addressed by combining P2x3, and Calb1 staining with a CT nerve label. The nerve label will allow the rostrocentral and rostromedial subdivisions to be identified in coronal sections and high magnification images from each of these regions may reveal Calb1 terminal staining in the rostromedial subdivision, but not in the rostral central subdivision. This would be a solid test of the author's hypothesis, and so should be included.

Minor Concerns:

1) Sentence structure throughout the paper should be reviewed. The first sentence in the introduction is a run-on. Clarity and concision improve impact.

2) On page 4, line 18, the cells were sequenced to a depth of 106 reads per cell, it would be useful to the reader to also know how this compares to references 9, 12, 15, and 16.

3) Hmx1 is a transcription factor that has been reported to regulate the development of the somatosensory portion of the geniculate (Quina et al., 2012). The authors should determine if this is one of the factors that define the black population in their study, which would support their data. Alternatively, if there is disagreement that might be interesting. Regardless, this paper should be considered.

4) Page 5 line 22, the authors state that Phox2b and Prrxl1 account for all neurons in the geniculate ganglion. That is a fairly global statement for an N of 2 where every other section is sampled. Revise to "all examined neurons".

5) On page 6 the authors mention immunohistochemistry data for Pou4f2, but the data aren't shown. The authors should either add the data to the supplementary materials or remove the sentence.

6) For identifying subtypes of neurons, the authors used various FPKM values to divide the neurons. Initially, they used values of greater than 200 because they only wanted to examine highly expressed genes, so an explanation was provided. However, later, for taste neuron subpopulations, they used greater than 5 times somatosensory values and for neurotransmitter subtypes greater than 30. It would be useful to the reader if explanations as to why these values were chosen were also provided.

7) Page 7, end of middle paragraph. The brightness or weakness of immunohistochemical staining is meaningless and could be due to a number of factors including background levels or the proximity of a nucleus to the cut surface of a section. The discussion of whether or not the nucleus was brightly or weakly stained might mistakenly give the impression that this matters to the interpretation of the data. Since it doesn't, remove information about degree of staining. Nuclei are either stained or unstained. Instead, report the FPKM values for Mafb for taste vs somatosensory neurons, which are the actual data indicating that expression is different in these two subpopulations. At these lower FPKM values, is there any evidence of Mafb expression in t1 or t3 taste populations? If not, Mafb still defines a taste subpopulation. The fact that some somatosensory neurons also express Mafb is also consistent with the t2 group being somatosensory in nature. Therefore this finding supports the author's conclusions. This should be mentioned here or in the later section.

8) On page 9 line 10, an experiment with 70 neurons is described as "preliminary", when the actual experiment has fewer neurons, it is unclear why this experiment is preliminary.

9) The "preliminary" experiment is not described in methods and it is unclear why higher concentrations of ATP and 5HT are used in this experiment. Also there is no description in the methods or data shown for what happens when calcium is removed from the bath. This entire experiment requires more explanation.

10) It would be useful for the reader to know how the concentrations of 5HT, ATP and GABA used here, compare with the literature (i.e. are they standard?) and a reference provided.

11) The scale in Figure 6f is hard to understand; the log $\Delta F/F$ is reported as -1 to 1. However, according to the methods, only responses that are ≥ 3 above baseline were counted, and it is unclear

how negative responses are accomplished from Figure 6e, so this scale is either wrong or it needs to be clarified.

12) In Figure 7b, because a subset of somatosensory neurons label with *Mafb*, it is unclear if the *Mafb*-positive cells are the t2 cells or somatosensory neurons without the addition of *Phox2b*, or both. This could be clarified with double-labeling with *Phox2b*.

13) In the discussion, the authors provide a thorough description of all possible meanings for the gustatory neuron subtypes, but some are more likely than others. Figure 8c shows *Mafb*-positive neurons in both CT and GSP labeled neurons. Therefore, data within this manuscript argue against this possibility; so this should be mentioned.

Breza JM, Travers SP (2016) P2X2 Receptor Terminal Field Demarcates a "Transition Zone" for Gustatory and Mechanosensory Processing in the Mouse Nucleus Tractus Solitarius. *Chemical senses* 41: 515-524.

Corson J, Aldridge A, Wilmoth K, Erisir A (2012) A survey of oral cavity afferents to the rat nucleus tractus solitarius. *J Comp Neurol* 520:495-527.

Quina LA, Tempest L, Hsu YW, Cox TC, Turner EE (2012) *Hmx1* is required for the normal development of somatosensory neurons in the geniculate ganglion. *Dev Biol* 365:152-163.

Substantive changes in the manuscript are identified by **blue font**. In addition to all points made by the Reviewers, we have supplemented the data. We added mice so that at least 3 mice were used for each quantified analysis in Results. We also conducted additional Ca²⁺ imaging; there are now twice as many neurons in Figure 6g; the outcome is unchanged.

Reviewer #1 major concerns:

1. The Reviewer makes an excellent point and we have followed this suggestion. PCA graphs indeed now show groups with more separation. For Figure 4b, we filtered out minimally-expressed genes by including only those with FPKM ≥ 10 in at least 3 taste neurons. This decreased the number of genes in the PCA from the original 17,225 down to the most informative 8150 genes. As the Reviewer predicted, this resulted in better separation between the taste sub-clusters. We have replaced the corresponding panel in Fig4b. We also applied the same suggestion to Figure 5a (PCA on transcription factor genes) to remove non-informative genes. The taste sub-clusters are slightly better separated.
2. We appreciate this point and have now explained the rationale for each of the filtering steps for selection of genes for HCA and PCA (Methods, p.19,20). The thresholds were not set based on particular “desired” groupings, but instead, to eliminate non-informative low-expressed genes (Fig.2a), or to emphasize taste-selective genes that might reveal taste neuron sub-types (Fig.4a).
3. We too had noticed the sub-subcluster within T1 that the Reviewer notes. As requested, we added this observation and one other obvious sub-cluster to the Discussion section (p.15).
4. We agree with the Reviewer that the where the potentially novel neuron type projects is intriguing. We carried out further immunohistochemical experiments and improved signal-to-noise. We now include new micrographs (Figure 8e,h,i) to suggest where the t2 neurons terminate in the NST. We have also included a detailed description in Results (p.12) and discussed how this terminal field compares to published findings with other electrophysiological and anatomical methods (p.15). These observations are intriguing and we are preparing more detailed tracing experiments for a forthcoming publication.

Reviewer #1 minor points:

1. We appreciate this point. However, there is a technical difficulty with implementing the suggestion. The Phox2b and Mafb antibodies used in other panels are both raised in goat and, separating the two signals was difficult. The one other anti-Phox2b antibody we tested is raised in mouse and gave a weaker signal with discernible background on mouse tissue. Although we have included a sample micrograph as Supplemental S6 (previously S5c), it just isn't possible to get consistent staining across multiple sections and confidently count incidence of co-expression with two antibodies from the same species. We do show that Phox2b and Prrxl1 are non-overlapping and account for all neurons in the ganglion (Figures 3b and 5g). Hence, the overlap of Mafb and Phox2b is readily inferred.
2. We agree with the Reviewer. We have quantified the GFP fluorescent intensity in *Htr3a*-GFP neurons. We show the quantification in Figure 7c,e, and describe on p.10 lines 25-29.
3. We agree. We have now explicitly stated this point on p.10 (lines 2-6).
4. Thank you for catching the deletion. We have added cyan cells to the legend (Figure 7a).
5. We have added the suggested phrase (p.10 line 32).
6. We removed the extra “neurons” from the sentence (now p. 11 line 25).

Reviewer #2 major concerns:

1. We have added a provisional statement of data availability (Methods, p.20). Data will be released upon publication. This is consistent with Nature journals' editorial office guidance.
2. We agree with this point, which was also Reviewer #1 point 4. As explained above, we have completed additional experiments, replaced micrographs in Figure 8, elaborated the description in Results (p. 12), and compared against published literature in Discussion (p.15). As mentioned above, we have begun more detailed analyses as a separate study, which will require some method development, optimization of labeling and immunostaining, replications, etc. Such data will be included in a forthcoming publication. We strongly believe that our substantial findings on sequence data from geniculate ganglion neurons cannot be held back awaiting the development of a second story. –

Reviewer #2 minor points:

1. We have edited the first sentence and throughout.
2. Ref.#9 lists 1.1×10^6 reads/cell; ref#15 is a review and lists publications that report 5×10^5 to 5×10^6 ; ref #16 lists several respected publications with reads per cell from 1.3×10^4 to 8.7×10^6 . As we stated, our values of $1-3 \times 10^6$ reads/cell (Supplemental Figure S3a) compare favorably.
3. We added a discussion (p.13 lines 18-22) of *Hmx1* in embryonic development of the ganglion. We also replaced one of the other genes with *Hmx1* in Figure 5a. In our adult data set, this gene is expressed at only low levels and only in a few somatosensory neurons.
4. We now have cell counts from 3 mice (p.5 lines 21-22) and changed the text accordingly.
5. We added the data requested on *Pou4f2* (Supplemental Figure S4d,e).
6. This same point was raised by Reviewer #1, point #2. We addressed this in Methods as stated above.
7. While immunofluorescence may not always be quantitative, in this case, it is. We document this in a new panel, Figure 5i, described on p.7 line 23-27. As shown in the heat map (Figure 5b), *Mafb* is undetectable in cluster t1 and t3 neurons.
8. We agree with the Reviewer on this point. We have now used the preliminary control experiment only to confirm the stability of the preparation (Supplemental Figure 8b,c). The main experimental data set on ATP, 5-HT and GABA is now supplemented with additional neurons from newly conducted experiments (p.9 last paragraph, Figure 6g). The outcome is unchanged.
9. As stated above (#8), we have removed the preliminary experiment. We have added a panel (now Figure 6e, and p.9 lines 19-22) to show the effect of eliminating bath Ca^{2+} .
10. We have justified (p.22, line 12-14) the concentrations of ATP, 5HT and GABA based on published pharmacology.
11. We have changed the scale on the heatmap (now Figure 6g) to more intuitively reflect GABA inhibition and described this in the Figure legend and in Methods (p.22).
12. We have clarified in Figure 7d (not 7b) that only the *Mafb*-bright neurons are visible without saturating the image. The suggestion to double label with *Phox2b* and *Mafb* is not technically feasible because both are rabbit antibodies, as we explain for Reviewer #1 point 1.
13. We do not understand the Reviewer's comment. In Figure 8c, there are only four *Mafb* nuclei in the field (arrowheads), and all four are in CT labelled (red) neurons. No *Mafb* nuclei are in GSP-labeled neurons.

REVIEWERS' COMMENTS:

Reviewer #1 (Remarks to the Author):

The authors have adequately addressed the concerns presented; the extra information regarding the selection of genes is excellent, as is the addition of experimental data to increase statistical power and dig deeper into the projection patterns of multiple subclasses of neurons.

Reviewer #2 (Remarks to the Author):

The manuscript is much improved and the reviewers have address all the major concerns. In particular the central nervous system anatomy data is clearer, and now adds to the manuscript.

With regard to my unclear comment concerning CT and GSP labels. The manuscript gives multiple explanations for what the differences in expression could mean. Including the possibility that the taste subclusters "represent input from the two distinct gustatory afferent nerves (CT, GSP) and the different receptive fields they innervate (fungiform taste buds of anterior tongue, foliate taste buds of lateral tongue, and taste buds of palate and naso-incisor duct)". The authors have data as to whether this is likely because they did CT and GSP labels and then labeled for Mafb. It might be mice to add one sentence stating whether these data are consistent or inconsistent with this possibility. I leave whether or not this should be added to the discretion of the authors and the editors.

Reviewers' second set of comments (NCOMMS-17-12283A, July, August 2017)

Reviewer #1: The authors have adequately addressed the concerns presented; the extra information regarding the selection of genes is excellent, as is the addition of experimental data to increase statistical power and dig deeper into the projection patterns of multiple subclasses of neurons.

Reviewer #2: The manuscript is much improved and the reviewers have address all the major concerns. In particular the central nervous system anatomy data is clearer, and now adds to the manuscript.

With regard to my unclear comment concerning CT and GSP labels. The manuscript gives multiple explanations for what the differences in expression could mean. Including the possibility that the taste sub-clusters “represent input from the two distinct gustatory afferent nerves (CT, GSP) and the different receptive fields they innervate (fungiform taste buds of anterior tongue, foliate taste buds of lateral tongue, and taste buds of palate and naso-incisor duct)”. The authors have data as to whether this is likely because they did CT and GSP labels and then labeled for Mafb. It might be mice to add one sentence stating whether these data are consistent or inconsistent with this possibility. I leave whether or not this should be added to the discretion of the authors and the editors.

Author Response to Reviewers' second set of comments

Reviewer # 1: Thank you for the compliments without concerns.

Reviewer # 2: We appreciate the Reviewer's interest in this discussion point. However, this is not the thrust of our manuscript, and our experiments were not designed to test this point. The very small size of the T2 cluster will require several experiments to establish whether a single or both nerves contain T2 neurons. It would be scientifically irresponsible to speculate beyond what we have done in the discussion.